# Symmetry breaking of tissue mechanics in wound induced hair follicle regeneration of laboratory and spiny mice

Hans I-Chen Harn [1,2], Sheng-Pei Wang [1,2], Yung-Chih Lai[3], Ben Van Handel[4], Ya-Chen Liang [1,3], Stephanie Tsai[1,5,6], Ina Maria Schiessl [7], Arijita Sarkar[4], Haibin Xi[8,9], Michael Hughes [2], Stefan Kaemmer [10], Ming-Jer Tang[2,11], Janos Peti-Peterdi [7], April D. Pyle [8,9,12,13], Thomas E. Woolley [14], Denis Evseenko[4,15], Ting-Xin Jiang[1] & Cheng-Ming Chuong[1✉]

Tissue regeneration is a process that recapitulates and restores organ structure and function. Although previous studies have demonstrated wound-induced hair neogenesis (WIHN) in laboratory mice (*Mus*), the regeneration is limited to the center of the wound unlike those observed in African spiny (*Acomys*) mice. Tissue mechanics have been implicated as an integral part of tissue morphogenesis. Here, we use the WIHN model to investigate the mechanical and molecular responses of laboratory and African spiny mice, and report these models demonstrate opposing trends in spatiotemporal morphogenetic field formation with association to wound stiffness landscapes. Transcriptome analysis and K14-Cre-Twist1 transgenic mice show the Twist1 pathway acts as a mediator for both epidermal-dermal interactions and a competence factor for periodic patterning, differing from those used in development. We propose a Turing model based on tissue stiffness that supports a two-scale tissue mechanics process: (1) establishing a morphogenetic field within the wound bed (mm scale) and (2) symmetry breaking of the epidermis and forming periodically arranged hair primordia within the morphogenetic field (μm scale). Thus, we delineate distinct chemo-mechanical events in building a Turing morphogenesis-competent field during WIHN of laboratory and African spiny mice and identify its evo-devo advantages with perspectives for regenerative medicine.

[1] Department of Pathology, Keck School of Medicine, University of Southern California, Los Angeles, CA, USA. [2] International Research Center of Wound Repair and Regeneration (iWRR), National Cheng Kung University, Tainan, Taiwan. [3] Integrative Stem Cell Center, China Medical University Hospital, China Medical University, Taichung, Taiwan. [4] Department of Orthopaedic Surgery, Keck School of Medicine, University of Southern California, Los Angeles, CA, USA. [5] Ostrow School of Dentistry, University of Southern California, Los Angeles, CA, USA. [6] School of Dentistry, National Taiwan University, Taipei, Taiwan. [7] Department of Physiology and Neuroscience, Zilkha Neurogenetic Institute, Keck School of Medicine, University of Southern California, Los Angeles, CA, USA. [8] Department of Microbiology, Immunology, and Molecular Genetics, University of California Los Angeles, Los Angeles, CA, USA. [9] Eli and Edythe Broad Center of Regenerative Medicine and Stem Cell Research, University of California Los Angeles, Los Angeles, CA, USA. [10] Park Systems Inc., 3040 Olcott Street, Santa Clara, CA 95054, USA. [11] Department of Physiology, Medical College, National Cheng Kung University, Tainan, Taiwan. [12] Molecular Biology Institute, University of California Los Angeles, Los Angeles, CA, USA. [13] Jonsson Comprehensive Cancer Center, University of California Los Angeles, Los Angeles, CA, USA. [14] Cardiff School of Mathematics, Cardiff University, Senghennydd Road, Cardiff, UK. [15] Department of Stem Cell Research and Regenerative Medicine, University of Southern California, Los Angeles, CA, USA. ✉email: cmchuong@med.usc.edu

The ultimate goal of regenerative medicine is to restore the function and structure of the original tissue. Wound healing in adult humans and mice generally undergoes re-epithelialization successfully yet fails to develop further, resulting in a scar with excess collagen and an absence of other skin appendages such as hair follicles. To facilitate regenerative wound healing, we look into skin development to recapitulate principles of hair follicular neogenesis.

Patterns form with the break of homogeneity and lead to the emergence of new structure or arrangement[1]. In skin development, Turing reaction-diffusion was shown to be involved in the periodic pattern formation of feathers and hairs[2,3]. Yet, before periodic patterning occurs, a morphogenetic field competent for Turing mechanism must take place; this should have proper cell density, the ability to secret morphogens, and appropriate morphogen receptors[4]. Within the morphogenetic field, FGFs, Wnt/β-catenin and Edar[5–7] signaling activate the epithelial cells to aggregate and form hair placodes, which later interact with the dermal condensate (DC) and invaginate into the dermis to form the foundation of a hair follicle. This process is characterized by a series of cohesive molecular signaling and also physical cellular events such as cell aggregation, collective cell migration and proliferation. Wnt/β-catenin signaling has been shown imperative to progress these cells into morphogenesis, in which a series of other signaling molecules such as Lef1, Sox2, Edar, Shh, MMP and Twist2 are also expressed[8,9].

Dynamic mechanical changes also occur during morphogenesis[10,11]. For epithelial cells to collectively migrate during morphogenesis, there must be an emergence of a local active stress acting at cell-cell or cell-matrix interfaces that creates an anisotropic force field[12–14]. Force generation by myosin-II motors on actin filaments have been shown to drives cell and tissue morphogenesis in drosophila embryonic development[15]. In other words, in order for hair placode to form and invaginate, the epithelial cells must overcome the physical barrier provided by the dermal cells and the extracellular matrix (ECM) in order to invaginate into the dermis. While previous studies uncovered mechanisms that can turn on/off hair follicle development, the chemo-mechanical dynamics that allows epithelial placode to form and invaginate into the dermis is largely unknown. Previously, Oster, Murray and Harris constructed a mathematical model that described the action of motile cells that could produce stress on their environment and thereby produce heterogeneous spatial patterns through mechanical means[16]. This basic theory assumes there are two populations: a motile population of cells, n, which can produce stress; and a tissue substratum, the ECM, which has density. Critically, the ECM is treated as a viscoelastic material, meaning that the ECM will deform subject to the traction forces produced by the cells. Critically, if the forces are added and removed quickly the ECM will relax back to its original shape. The convergence of these morphological and molecular asymmetries lead to the formation of DC[17] and activation of β-catenin in the adjacent epidermal cells to initiate feather bud gene expression[18,19].

In the avian skin development, the early formation of the morphogenetic field is recognized as the feather tract field, and individual buds form sequentially or simultaneously within the tract field, with some species-based differences[4,18,20]. This implies there are different ways to make competent morphogenetic fields[21], while the region outside of the tract field becomes the apteric region. Thus, in skin development, periodic generation of skin appendages occur in two steps: first the formation of morphogenetic field and then the periodic patterning of cell collectives within the field.

Wound-induced hair follicle neogenesis (WIHN) is a regenerative outcome of wound healing where fully function hair follicles develop de novo from the center of large full thickness excisional wounds[22]. This observation was originally described in rats, rabbits, sheep and even humans, and investigated in depth in recent years[23–26]. Cells in wound bed may use different paths to reach the morphogenetic competent state. We contemplate that to regenerate new hair follicles in the wound bed of the adult skin, a morphogenetic field also has to be established first, which then allows periodically arranged hair germs to be generated. Since adult cells have different epigenetic landscapes from embryonic cells, the generation of hair placodes in the adult may not follow the exact pathway as in development. Investigating the chemo-mechanical dynamics of the epithelial cells and the ECM during hair follicular neogenesis would facilitate our understanding on how to set up the morphogenetic field and initiate the signaling events, which would have translational potential.

In search for the ultimate regeneration model, the African spiny mouse (Acomys cahirinus) serves as an adequate model to study complete skin regeneration. The spiny mouse has evolved to give away up to 70% of its skin to its predator and still remarkably regenerates its entire skin and appendages[27–29]. Its skin is mechanically softer (20 times) than the laboratory mouse (Mus musculus) and much easier to break (77 times less energy required)[27]. The gene expression profiles of the spiny mouse wound show a dampened response of collagen, MMP and inflammatory molecules after wounding, suggesting an alternative microenvironment to enrich hair neogenesis[30]. On the other hand, studies in laboratory mice have shown that traction, or tension across the skin or wound, causes hypertrophic scar through FAK signaling[31], implying the significance of tensile state of the connective tissue to tissue functioning[32,33]. Nevertheless, the initiating events and how these findings can translate into our understanding in laboratory mice and human remain to be explored. Thus we hypothesize that spatial tissue mechanics of the wound partake in establishing the morphogenetic field for hair follicular neogenesis, and there lies a chemo-mechanical signaling event that initiates the symmetry breaking of the epidermis and leading to placode formation and invagination. We show Twist 1 pathway plays a key role in modulating tissue stiffness and facilitate hair formation. Furthermore, by delineating the common and distinct features of laboratory and spiny mouse during WIHN, we learn from evo-devo advantages to provide perspective for future implications.

## Results

**Tissue mechanics set up morphogenetic field for wound-induced hair neogenesis.** To examine the effects of tissue mechanics on wound induced hair neogenesis, we first created $1 \times 1$ cm full thickness wounds on the dorsal skin of 3-week-old C57Bl/6 mice, and observed new hair follicles formed at the center of the wound on post-wound day 28 (PWD28, Fig. 1a–c). To investigate the spatial stiffness distribution of the wound bed, we used an atomic force microscopy (AFM)[34] to measure across the wound (Supplementary Fig. S1a), and calculated tissue stiffness from force–displacement curves (Supplementary Fig. S1b, c) using a modified Hertz model[35,36]. We found that the center of the PWD14 wound, where de novo hair follicles can be observed, is significantly softer than the wound periphery ($28.0 \pm 1.1$ vs $10.5 \pm 0.6$ kPa, Fig. 1d, e).

To evaluate whether the observed tissue stiffness plays a role in hair follicle neogenesis, we treated the wound daily from PWD10 to PWD16 with 100 μM of Blebbistatin (Fig. 1f), and observed an increase in the number of hair neogenesis (Fig. 1g, h) and decrease in the stiffness of the wound bed (Fig. 1i). Statistically, Blebbistatin significantly increased the number of resultant hair follicles from $17.4 \pm 2.1$ to $34.5 \pm 4.3$ kPa (Fig. 1j, $n = 8$), and

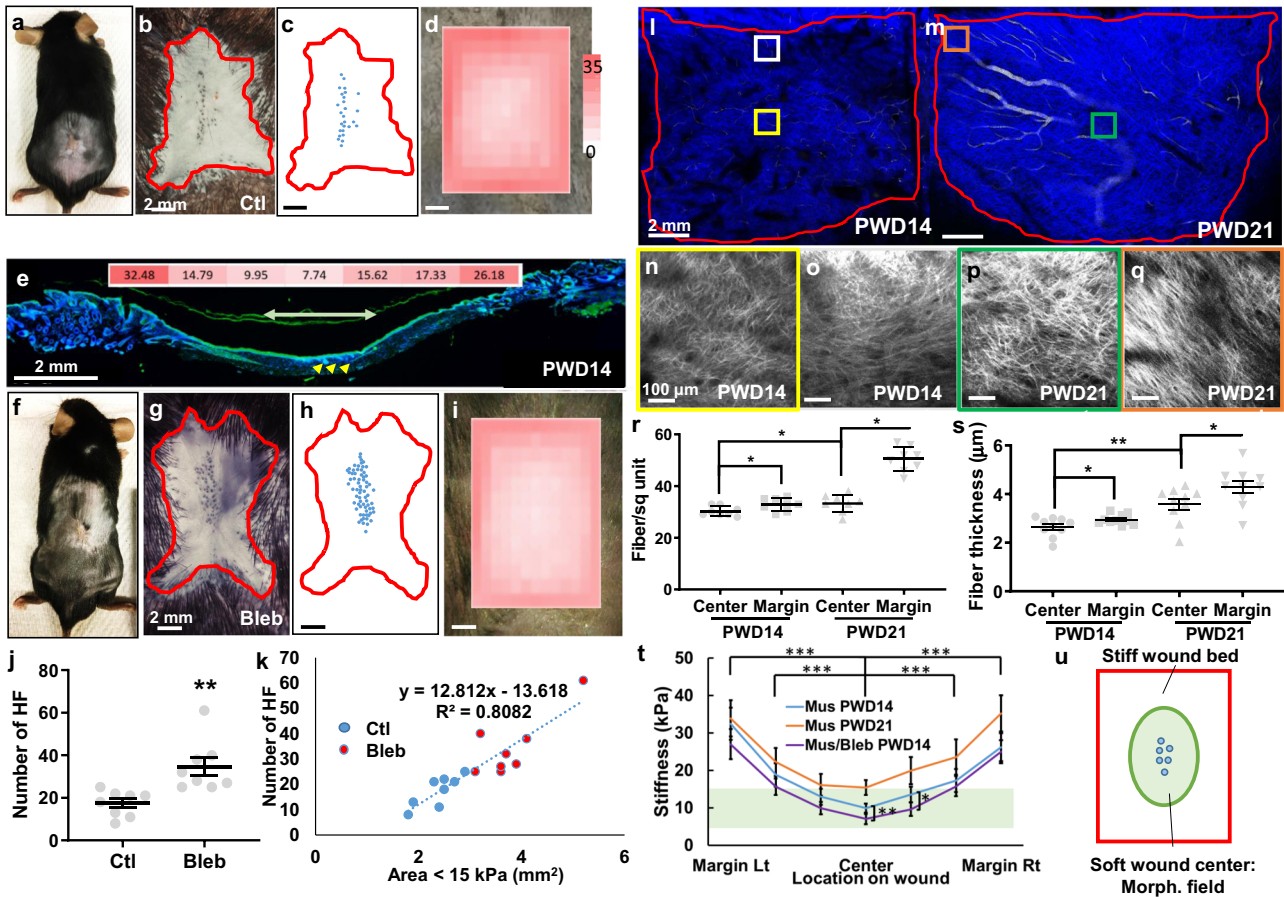

**Fig. 1 Tissue mechanics set up morphogenetic field for wound-induced hair follicle neogenesis. a** A PWD28 C57Bl/6 mouse. **b** AP staining showing de novo hair follicle formation at the center of the wound bed at PWD28. **c** Schematic diagram showing the location of regenerated hair follicles (blue dots) in **b**. **d** Stiffness heatmap overlaying the PWD14 wound. Colorimetric unit: kPa. **e** Cross-sectional view of the PWD14 wound and distribution of wound stiffness. Yellow arrow heads indicate the formation of hair placodes. Heatmap and number indicate the spatial stiffness of the whole wound bed. Unit: kPa. Green arrow indicates the range of the morphogenetic field. **f** A Bleb-treated mouse on PWD28. **g** AP stain showing the number of hair follicles increased significantly upon Bleb treatment. **h** Schematic diagram of hair follicles (blue dots) in **g**. **i** Stiffness heatmap of PWD14 Bleb-treated overlaying wound. **j** Dot plot showing changes in HF upon Bleb treatment. $n = 8$ biologically independent animals. Data are presented as mean values ± SEM. $p = 0.003$, unpaired two-sided $t$-test. **k** The area of the wound bed under 15 kPa positively correlates with the number of hair follicles. SHG of **l** PWD14 and **m** PWD21 wounds. The color squares indicate the location of corresponding enlarged photos from **n** PWD14 center, **o** PWD14 margin, **p** PWD21 center and **q** PWD21 margin of the wound image. **r** Dot plot showing the number of fibers per square unit. PWD14 C/M: $p = 0.0323$, PWD14/PWD21 C: $p = 0.0495$, PWD21 C/M: $p < 0.0001$. **s** Fiber thickness in respective wound time and location. PWD14 C/M: $p = 0.0454$, PWD14/PWD21 C: $p = 0.0020$, PWD21 C/M: $p < 0.0461$. **r**, **s** Data are presented as mean values ± SD. $n = 8$ regions examined over four biologically independent animals. One-way ANOVA, Tukey test. **t** Summary graph of wound stiffness on PWD14 and PWD21 with respect to wound location and Bleb treatment. $n = 10$ regions examined over five biologically independent animals per location per condition. Data are presented as mean values ± SD. *$p = 0.0478$; **$p = 0.0061$; ***$p < 0.0001$. One-way ANOVA, Tukey test. **u** Illustration of tissue mechanics partake in setting up the morphogenetic field for WIHN. Red line: wound margin. Green: soft, morphogenetic field. Blue: hair placodes. Ctl: control; Bleb: Blebbistatin; HF: hair follicle; sq: square; Lt: left; Rt: right; Morph: morphogenetic. The images in **a**–**i** represent 8 out of 8 experiments performed. The images in **l**–**q** represent 4 out of 4 experiments performed.

significantly increased the area of the wound bed that is under 15 kPa from $2.38 ± 0.13$ to $3.70 ± 0.26$ mm$^2$ (Supplementary Fig. S1d). To correlate tissue stiffness with hair neogenesis, we quantified and plotted the area of the wound under 15 kPa versus the number of hair neogenesis observed in each wound, and found a strong correlation ($R^2 = 0.8082$), suggesting stiffness may contribute to creating the morphogenetic field for hair neogenesis (Fig. 1k).

We further explored the molecular constituents of the wound stiffness. Collagen has been implicated as the main ECM of the wound[37], hence we used second harmonic generation (SHG) to visualize the amount and organization of collagen fibrils in the wound (Fig. 1l–q). Generally speaking, fibers in PWD14 wounds are thinner and less dense than in PWD21 ones. Both the fiber density

and thickness are significantly higher in the wound margin than in the wound center in PWD14 and PWD21 (Fig. 1r, s). Hence, the spatiotemporal organization of collagen fibrils in the PWD14 and PWD21 wound beds corresponds to its respective stiffness of the wound (Fig. 1t). This implies the wound bed needs to be softer than 15 kPa for hair neogenesis to occur (green bar, Fig. 1t). Blebbistatin treatment not only softened the entire wound bed, but also more importantly, significantly lowered the stiffness of intermediate margin-center region of the wound (blue and purple lines, Fig. 1t), hence setting up a larger morphogenetic field competent for hair neogenesis (green bar, Fig. 1u). These results suggest tissue mechanics play an important role in hair neogenesis, and the spatial organization of collagen fibrils may be the main constituent of tissue stiffness in the wound bed.

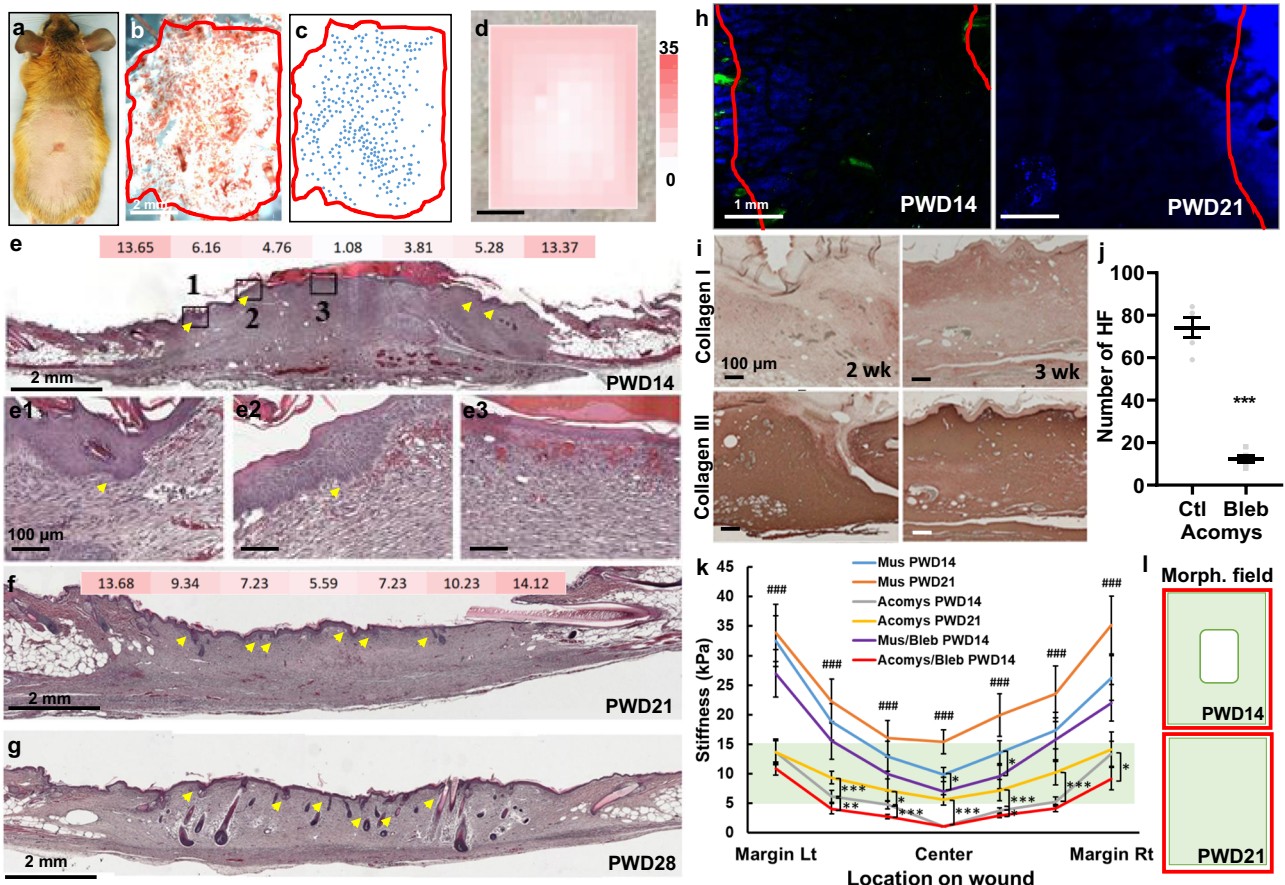

**Fig. 2 African spiny mice exhibit an optimal range of tissue stiffness for placode formation. a** A spiny mouse on PWD28. **b** K15 wholemount immunostaining of a PWD28 spiny mouse wound. Red line: border of the wound bed. **c** Schematic diagram of hair follicles (blue dot) in **b**. **d** The stiffness heatmap overlaying the PWD14 spiny mouse wound. Colorimetric unit: kPa. **e** H&E histology and stiffness heatmap of PWD14 and **f** PWD21 wounds. The color and number indicate the stiffness of the wound at respective location. Unit: kPa. (**e1–e3**) enlarged images of regions 1, 2, and 3 from **e**. **g** H&E of the PWD28 wound. Yellow arrows indicate the formation of hair placode. **h** SHG of PWD14 and PWD21 spiny mouse wounds. Red line: wound border. **i**, IHC of collagen I and collagen III in spiny mouse at different post-wound time. **j** Blebbistatin treatment significantly reduced the resultant number of hair neogenesis. $n = 5$ biologically independent animals. Data are presented as mean values ± SEM. $p < 0.0001$, unpaired two-sided $t$-test. **k** Graph indicating the respective wound stiffness of the specific wound location and time in the laboratory and spiny mice. $n = 10$ regions examined over 5 biologically independent animals per location per condition. Data are presented as mean values ± SD. $^{\#\#\#}p < 0.0001$ when compared between Mus PWD14 and Acomys PWD14; $^{*}p = 0.0482$ when compared between Mus PWD14 and Mus/Bleb PWD14 center, $p = 0.0218$ when compared between Mus PWD14 and Mus/Bleb PWD14 at the right of wound center, $p = 0.0441$ between Acomys PWD14 and Acomys PWD21 at the left of wound center, $p = 0.0381$ when compared between Acomys PWD14 and Acomys/Bleb PWD14 wound margin right, $p = 0.0421$ at center-right, $^{**}p = 0.0089$; $^{***}p < 0.0001$. One-way ANOVA, Tukey test. **l** Illustration of morphogenetic field in PWD14 and PWD21 spiny mouse wound bed. Red: wound border. Green: morphogenetic field. $^{\#\#\#}$: $p < 0.005$. Ctl: control; Bleb: Blebbistatin; Lt: left; Rt: right; Morph: morphogenetic. The images in **a–g**, **i** represent five out of five experiments performed. The images in **h** represent 4 out of 4 experiments performed.

**African spiny mice exhibit an optimal range of tissue stiffness for placode formation**. The African spiny mice are known to have robust ability in WIHN (Fig. 2a). Unlike laboratory mice that show de novo hairs only in the center of the wound bed, new hair follicles were observed across the entire wound bed on PWD28 (Fig. 2b, c). To explore how these events occur over time, we examined the spatiotemporal pattern of hair placode emergence in spiny mice. Interestingly, we found an opposite trend. In spiny mice, the hair placodes started to develop from the periphery of the wound bed on PWD14 (Figs. 2e, e1–3). The center of the wound bed did not form hair placodes until PWD21 (Fig. 2f). How can we explain this opposite trend? Can tissue mechanics play a role?

We used an AFM to determine the spatiotemporal dynamics of tissue stiffness during WIHN in spiny mice (Fig. 2d). In general, two trends are similar to that of the laboratory mice: (1) the center of the wound bed is softer than the wound periphery

(Fig. 2d), and (2) the overall stiffness of the wound increased from PWD14 to PWD21 (Fig. 2e, f). However, there are two major differences: (1) the soft nature of the unwounded spiny mice skin and wound bed, and (2) the periphery-to-center formation pattern of de novo hair primordia in the spiny mice wounds. The unwounded spiny mouse skin and the stiffest region of its wound bed, the wound margin, are still softer than 15 kPa, while the center of the wound bed is below 5 kPa on PWD14 (Fig. 2e). As a result, the hair primordia formation in the spiny mouse wound followed a periphery to central trend; the hair placodes only began to form where the wound stiffness was higher than 5 kPa (Fig. 2f). Hence, the de novo hair follicles were more mature at the wound periphery than at the wound center, and this morphological feature is apparent on PWD28 (Fig. 2g). This result is best explained by the presence of an optimal range between 5 and 15 kPa of tissue stiffness for placode formation. To test this hypothesis, we further softened the spiny mouse wound

bed by Blebbistatin treatment and showed that: (1) Blebbistatin treatment softened the wound bed and significantly increased the wound area under 5 kPa (Supplementary Fig. S2a–c), and (2) the resultant number of hair neogenesis was significantly reduced from 74.00 ± 6.14 to 12.40 ± 1.10 (Fig. 2j, Supplementary Fig. S2d). These results suggest that regions of the wound bed that are either too stiff or too soft are not favorable for new hair formation.

What is the molecular basis of these differences in tissue stiffness? To investigate the structure and organization of collagen fibrils in the spiny mouse wound as observed in the laboratory mice, we also used SHG to visualize it. Interestingly, collagen fibrils were identified in the unwounded spiny mouse skin but were almost undetectable within the wound center in both PWD14 and PWD21 wounds (Fig. 2h). We analyzed and compared the fiber thickness and fiber density of PWD14 and PWD21 spiny mouse wound margin (Supplementary Fig. S2e, f), and showed that the collagen fibrils in the spiny mouse wounds are significantly thinner and less dense than that of the laboratory mice (Supplementary Fig. S2g, h). The spiny mouse skin has been reported to have high levels of collagen III[27]; however, since type III are less crystallined and generate little SHG signal[38,39], we further used IHC to examine the expression of collagen I and III in the spiny mouse wounds. We found that collagen III is the dominant collagen type expressed in the early stages of spiny mouse wound healing (Fig. 2i); nevertheless, this relationship is reversed later in post 16-week wounds when collagen I was highly expressed in contrast to collagen III (Supplementary Fig. S2i). To summarize, we found that there is a lower limit on the softness of the wound bed at 5 kPa in order for hair neogenesis to occur, as the initial hair placodes can only be observed in wound stiffness between 5 and 15 kPa in both laboratory and spiny mice wounds (Fig. 2k). This implies that 5–15 kPa could be the optimal range for the wound to set up the morphogenetic field for hair placode formation, and suggests why hair neogenesis begins from the wound periphery and later to center in the spiny mice (Fig. 2l).

**RNA-seq analysis identifies epidermal Twist1 as an upstream regulator for the formation of hair placodes in WIHN**. To delineate the molecular mechanism underlining WIHN, specifically at the morphogenetic field, we used a 5 mm punch biopsy to separate the PWD14 epidermis into the regenerative wound center and the non-regenerative wound margin, and performed RNA-seq analysis to look for differentially expressed genes (DEG) between the two. PWD14 was selected because our preliminary data showed that gene expression related to hair neogenesis (*Wnt5a*, *Lef1*, *Gli1*, *Fgf10,* and *Twist1*) peaked in the wound center of PWD14 epidermis (Supplementary Fig. 3a, b). From this RNA-seq analysis, we identified 2780 DEG (Fig. 3a). Among them, Wnt/β-catenin signaling ($P = 1.49 \times 10^{-2}$), Integrin signaling ($1.84 \times 10^{-8}$), Stat3 pathway ($1.01 \times 10^{-6}$), inhibition of Matrix Metalloproteases ($4.53 \times 10^{-7}$), EMT core genes ($4.23 \times 10^{-10}$), proliferation of epithelial cells ($3.47 \times 10^{-17}$), cell movement of epithelial cells ($5.43 \times 10^{-13}$), and organization of ECM are significantly enriched (Fig. 3b, c, Table 1). Furthermore, we identify 20 significantly upregulated genes related to hair placode formation (Table 2).

To identify the potential upstream initiators of WIHN, we identified 114 transcription factors (TF) that are significantly upregulated at the wound center, including those that are related to the Twist (*Twist1*, *Twist2*, *Snai1*) and Wnt (*Tcf23*) pathways (Table 3). Among them, Twist1 is the second most statistically significant upstream regulator, which regulates many downstream DEG ($P = 4.55 \times 10^{-11}$) as indicated by the Upstream Analysis function in Ingenuity Pathway Analysis (IPA) (Fig. 3d, e). The

top significant transcription factor is *Nfkbia* ($3.28 \times 10^{-13}$) which is associated with the inflammatory response after wounding (Fig. 3d). Wholemount immunostaining also showed that epidermal Twist1 is highly expressed in the PWD14 wound center and enriched in both hair placode and inter-placode region more than in the margin (Fig. 3f, Supplementary Fig. 3c). IHC staining of Twist1 and Snai1 (number 5 ranked TF) on PWD14 sections show both TF are expressed in the epithelial hair placode (Fig. 3g). These findings point to ECM remodeling, cell proliferation, cell movement, Stat3, and Wnt/β-catenin as important signaling events during WIHN, and epidermal Twist1 appears to be an important upstream regulator for hair placode formation.

Given the differences between laboratory and spiny mice during WIHN[29] (Figs. 1, 2), we also performed RNA-seq analysis on spiny mouse wounds to identify key molecules for its hair primordia formation. Since the entire spiny mouse wound bed is capable of undergoing WIHN, we harvested the entire wound on PWD0, 14, 21, and 28 and separated them into epidermis and dermis to analyze the quantitative changes of gene expression before, and during early, mid, and late stages of WIHN (Fig. 4a). From the wound epidermis, the proliferation of epithelial cells and cell movement associated genes all peaked on PWD14, including *Twist1* (Fig. 4b). In the dermis, *Twist1* and its related TF *Zeb2* also peaked on PWD14, while *Zeb1* and *Tgfb1* showed a different trend (Fig. 4c). Interestingly, many of the ECM and MMP related genes along with other TFs such as *Stat3* are highly expressed on PWD21 and PWD28 (Fig. 4d), which may reflect the softness of PWD14 spiny mouse wounds (Fig. 2k). IHC staining of PWD14 spiny mouse wounds also show that Twist1, Zeb2, MMP9 and P-cad are expressed around the hair placode, while E-cadherin is downregulated at the hair placode. As suggested by its RNA-seq analysis, Twist1, Zeb2 and MMP9 are also expressed in the dermis (Fig. 4e). Lastly, using Harmine and GM6001, we show that inhibiting Twist1 and pan-MMP activity in the spiny mouse wounds also significantly reduced WIHN (Fig. 4f, g).

It should be noted that while dermal Twist1 is highly expressed in embryonic skin, it is absent in the E14 epidermal hair placode (Supplementary Fig. S3d). Identifying epidermal Twist1 in WIHN suggests that the spatiotemporal dynamics of the tissue stiffness and gene expression may play a more important role in the epithelial placode formation in WIHN than in embryonic development.

**Epidermal Twist1 regulates both epidermal and dermal cell behavior and tissue stiffness towards hair primordia formation in WIHN**. To verify the role of epidermal Twist1 in WIHN, we crossed the K14-Cre mice with Twist1-loxP mice to generate the K14-Cre-Twist1 mutant mice (Fig. 5a) that expressed little Twist1 in the epidermis (Supplementary Fig. S4a). The mutant mice showed a significant decrease in hair neogenesis (Fig. 5b, c, n), and a significantly stiffer wound bed than that of wild type mice (Fig. 5d, o, Supplementary Fig. S4b). We further used small molecule inhibitors to suppress Twist1 (Harmine, Supplementary Fig. S4a) and its downstream MMP activities (GM6001), and both treatments significantly decreased the number of hair neogenesis (Fig. 5e–j, n), stiffened the wound center (Fig. 5o, Supplementary Fig. S4b), and reduced the wound area under 15 kPa (Supplementary Fig. S4c). Interestingly, in the perturbed wounds, the wound area under 15 kPa showed a correlation with the resultant number of hair follicles ($R^2 = 0.7496$, Supplementary Fig. S4d), although the slope of the trend line is much smaller than that of the control and Blebbistatin treated samples (1.6828 vs 12.812, Supplementary Fig. S4d and Fig. 1k).

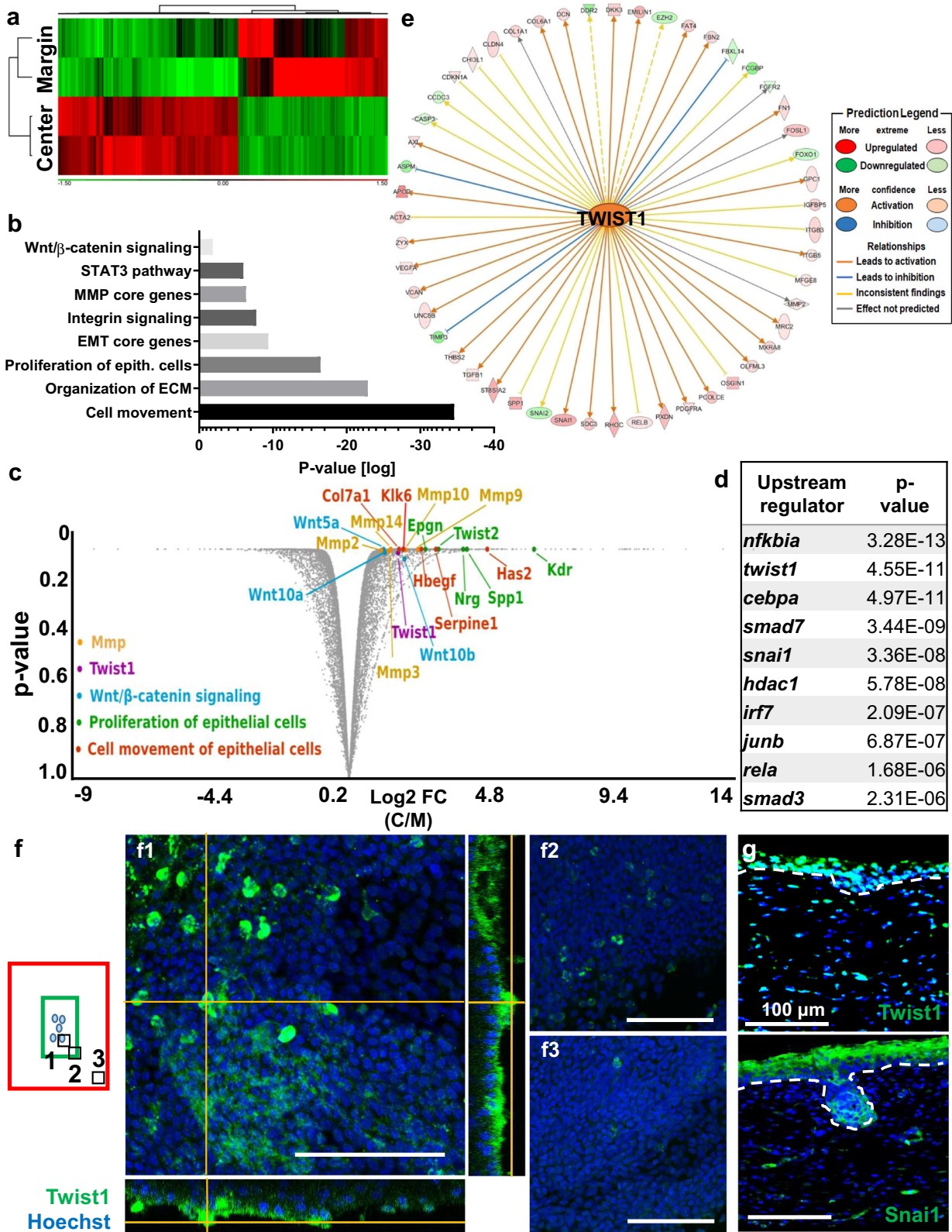

We further used lentivirus to transfect and overexpress Twist1 in the wild-type mouse wounds on PWD10 and observed a significant increase in the resultant hair follicle numbers on PWD28 (Fig. 5k–n). Correspondingly, we re-analyzed public microarray data that compared mouse strains with high and low WIHN capacity[40], and found that Twist1 expression levels are also significantly higher in the high regenerative strain (C57BL/6 X FVB X SJL) than the low regenerative one (C57BL/6, $p = 0.004$, Supplementary Fig. S4e). These findings verify that Twist1 and its downstream signals such as MMP play important roles in controlling wound stiffness and hair follicle neogenesis during WIHN (Fig. 5n, o).

**Fig. 3 Transcriptome analysis identifies Twist1 as an important transcription factor during WIHN in laboratory mice. a** Gene expression heatmap of PWD14 epidermis wound center vs wound margin. **b** The significantly enriched pathways of DEG. **c**, Volcano plot showing gene expression fold changes of representative DEG in MMP, Twist1, Wnt/β-catenin related pathways, proliferation of epithelial cells, cell movement of epithelial cells. Upregulation indicates high expression in wound center. FC: fold change. C/M: wound center versus margin. **d** Twist1 ranked 2nd by two-sided Fisher's Exact Test $p$ value in the top ten differentially expressed upstream regulator. **e** IPA identifies Twist1 as the top upstream regulator of the downstream DEG ($p = 1.1 \times 10^{-21}$). **f** Wholemount immunostaining of Twist1 at respective location from the epidermal side of the PWD14 wound as illustrated. Red line: wound border. Green line: morphogenetic zone. Blue dots: hair placode. **g** IHC of Twist1 and Snai1 in PWD14 wounds containing the hair placode. Dotted line demarcates the border of dermal epidermal junction. Blue: Hoechst. Scale bar: 100 μm. **a–d** $n = 2$ biologically independent experiments. The images in **f**, **g** represent 4 out of 4 biologically independent experiments performed.

---

**Table 1 The significantly enriched pathways in PWD14 epidermis wound center vs margin RNA-seq analysis.**

| Enriched Pathway | p-value | Differentially expressed genes |
|---|---|---|
| Wnt/β-catenin signaling (X30) | $1.49 \times 10^{-2}$ | appl2, bmpr2, cdh5, csnk1d, dkk3, dvl1, ep300, fzd2, gsk3b, hdac1, kremen2, lrp1, lrp6, ppard, ppp2r1a, ppp2r2c, ppp2r3a, ppp2r5b, ptpa, sfrp1, sox15, sox18, src, tgfb1, tgfb3, tle3, wnt16, wnt10a, wnt10b, wnt5a |
| Regulation of EMT Pathway (X28) | $1.02 \times 10^{-2}$ | dvl1, fzd2, hgf, lox, map2k2, mmp2, mmp9, notch4, pard6b, pdgfrb, ralb, smad3, smurf1, snai1, tgfb1, tgfb3, twist1, twist2, wnt10a, wnt10b, wnt5a, wnt9b |
| MMP genes (X1) | $4.53 \times 10^{-7}$ | adam12, hspg2, lrp1, mmp2, mmp3, mmp9, mmp10, mmp11, mmp13, mmp14, mmp17, mmp19, mmp23b, sdc1, thbs2, timp1, timp3 |
| Organization of ECM (X84) | $1.32 \times 10^{-23}$ | adam12, adamts4, agrn, apbb2, aplp1, bgn, bmp1, c6orf15, col16a1, col18a1, col1a1, col1a2, col27a1, col3a1, col4a1, col4a2, col5a1, col5a2, col5a3, col6a1, col6a2, col6a3, col6a4, col7a1, col8a1, ctsk, dcn, ddr1, eglflam, elf3, emilin1, fbln5, fbn1, fbn2, fn1, furin, icam2, ibsp, itga1, itga11, itga3, itga5, itga9, itgb3, itgb6, jam2, kdr, lama4, lox, loxl1, mfap2, mmp1, mmp10, mmp11, mmp13, mmp14, mmp19, mmp2, mmp9, nid1, nid2, olfml2a, olfml2b, pdgfra, pecam1, postn, prdx4, ptx3, pxdn, serpine1, sh3pxd2b, sparc, spp1, timp1, tnc, tnf, vcam1, vcan, vtn, vwf |
| Proliferation of epithelial cells (X186) | $3.47 \times 10^{-17}$ | ahr, alms1, areg, atm, bad, bcl11b, birc2, bnc1, brca1, calm1, casp3, casp8, ccnd3, cd9, cdc25b, cdc73, cdkn1a, cdkn1b, cebpa, cers2, col8a1, creb3l3, cryab, csf2rb, ctsv, cul3, cxcr2, dab2ip, eif4e, eng, ep300, epgn, epha2, ercc1, ereg, esrra, ezh2, fbln5, fgfr2, fn1, frs2, fst, gata3, glul, grn, gsk3b, has2, hbegf, heyl, hgf, hoxa5, hyal1, ifngr1, ift52, ift74, ift80, igf1, il18, il22ra2, il24, il4r, il6r, inhba, inhbb, itga1, itga3, itgb3, junb, kcnk2, kdr, klf10, klf5, klk3, klk6, klk8, krt16, krt17, lgals7/lgals7b, lgr4, lmnb1, lrp6, maged1, map2k1, map2k6, map2k7, mapk7, mapk8, mapk9, mapkapk2, marveld3, mfge8, mki67, mmp14, mmp9, mt2, nab1, nab2, nfib, nfkbia, nme2, npm1, nr3c1, nrg1, odc1, p2rx7, pkp3, postn, ppard, prlr, ptafr, pten, ptgs2, pthlh, ptpn1, ptprk, rack1, rbl2, rela, relb, rgn, rida, rps6kb1, s1pr2, sema4d, serpinf1, sfn, sfrp1, sh2b1, slc20a1, slc7a5, smad3, smad7, snai2, socs1, socs3, sparc, spint2, spp1, stat5a, stmn1, tfrc, tgfb1, tgfb3, tgm1, timeless, timp1, tnfaip6, tnfrsf11a, tnfrsf12a, tnfrsf1a, tslp, twist1, twist2, uhrf1, vegfa, wnt10b, wnt16, wnt5a, yod1, zbtb16 |

Comparison using two-sided Fisher's Exact Test.

---

**Table 2 The 20 significantly upregulated genes related to hair placode formation at epidermis wound center compared to wound margin on PWD14.**

| Gene | Description | Location | Type(s) |
|---|---|---|---|
| cdh11 | cadherin 11 | Plasma Membrane | other |
| meg3 | maternally expressed 3 | Other | other |
| twist1 | twist family bHLH transcription factor 1 | Nucleus | transcription regulator |
| col1a1 | collagen, type I, alpha 1 | Extracellular Space | other |
| col3a1 | collagen, type III, alpha 1 | Extracellular Space | other |
| col6a2 | collagen, type VI, alpha 2 | Extracellular Space | other |
| col6a3 | collagen, type VI, alpha 3 | Extracellular Space | other |
| efemp2 | EGF containing fibulin-like extracellular matrix protein 2 | Extracellular Space | other |
| fbn1 | fibrillin 1 | Extracellular Space | other |
| fstl1 | follistatin like 1 | Extracellular Space | other |
| lgals1 | lectin, galactoside-binding, soluble, 1 | Extracellular Space | other |
| timp2 | TIMP metallopeptidase inhibitor 2 | Extracellular Space | other |
| cpxm1 | carboxypeptidase X (M14 family), member 1 | Extracellular Space | peptidase |
| dpysl3 | dihydropyrimidinase like 3 | Cytoplasm | enzyme |
| ftl1 | ferritin, light polypeptide | Cytoplasm | enzyme |
| pde5a | phosphodiesterase 5A | Cytoplasm | enzyme |
| ppic | peptidylprolyl isomerase C | Cytoplasm | enzyme |
| myh10 | myosin, heavy chain 10, non-muscle | Cytoplasm | other |
| srpx2 | sushi-repeat containing protein, X-linked 2 | Cytoplasm | other |
| igf2bp2 | insulin like growth factor 2 mRNA binding protein 2 | Cytoplasm | translation regulator |

**Table 3 The 114 significantly upregulated transcription factors at epidermis wound center compared to wound margin on PWD14.**

| | | | | | | | |
|---|---|---|---|---|---|---|---|
| acap3 | cers2 | fam129b | hr | mier2 | relb | srebf1 | wtip |
| aes | churc1 | fem1a | id3 | mllt1 | rfx1 | srebf2 | zbtb42 |
| arid3a | cic | fiz1 | ier2 | mnt | rfxank | srf | zdhhc13 |
| arid5a | cited4 | fosl1 | ifi204 | mxd1 | rnf114 | ssbp4 | zfp219 |
| asb1 | creb3l3 | fosl2 | irf1 | nab2 | rnf25 | tbx15 | zfp369 |
| asb6 | csrnp1 | foxc1 | irf5 | nfatc4 | rnf4 | tbx3 | zfp444 |
| atf4 | ctbp2 | foxp4 | irf7 | nfkbia | sbno2 | tcf23 | zfp593 |
| atf6b | dnmt3l | glis2 | irx1 | noct | scand1 | tcfl5 | zkscan6 |
| atxn7l3 | e2f7 | glmp | jarid2 | notch4 | siah2 | thap4 | zxdc |
| barx2 | eaf1 | gpank1 | junb | pax9 | smad3 | trim16 | |
| bhlhe40 | ehmt2 | hdac1 | klf10 | phf1 | smad7 | tsc22d1 | |
| bnc1 | elf3 | helz2 | lmo1 | pitx1 | snai1 | tsc22d4 | |
| btg2 | elk1 | heyl | lztr1 | prrx1 | sox15 | twist1 | |
| carhsp1 | elk3 | hic2 | maff | rbpms | sox18 | twist2 | |
| cebpa | esrra | hopx | maged1 | rela | sqstm1 | ube2v1 | |

Based on p-value < 0.05.

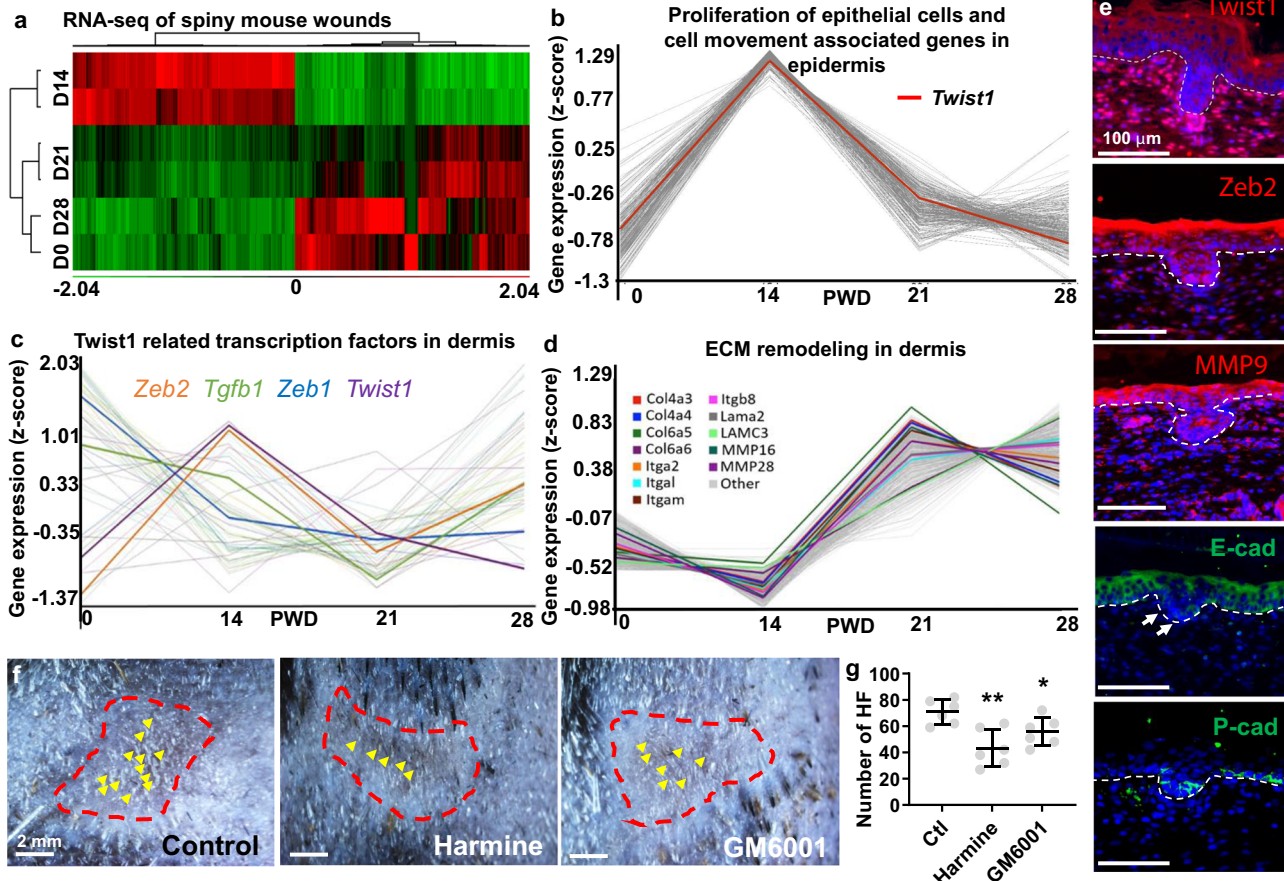

**Fig. 4 Twist1 is also expressed and can modulate the outcome of WIHN in spiny mice. a** Gene expression heatmap of spiny mouse epidermis at different post-wound days. **b** Proliferation of epithelial cells and cell movement associated genes in the spiny mouse wound epidermis. **c** Twist1 related transcription factors in the spiny mouse wound dermis. **d** ECM remodeling in the spiny mouse wound dermis. **e** IHC of Twist1, Zeb2, MMP9, E-cad and P-cad in PWD14 the regenerating hair placodes of the spiny mice. White arrows point to the downregulation of E-cad at the tip of hair placode. **f** The effects of Twist1 inhibitor Harmine and pan-MMP inhibitor GM6001 treatment on hair neogenesis in spiny mouse. Red: wound border; yellow arrow: hair follicles. Observed on PWD35. **g** Dot graph of the number of regenerated hair follicles (HF) observed in spiny mouse wounds under different treatments. $n = 6$ biologically independent animals. Data are presented as mean values ± SEM. *$p = 0.0265$; **$p = 0.0027$; unpaired two-sided t-test. Ctl: control. The images in **f** represent six out of six experiments performed. The images in **e** represent 4 out of 4 experiments performed.

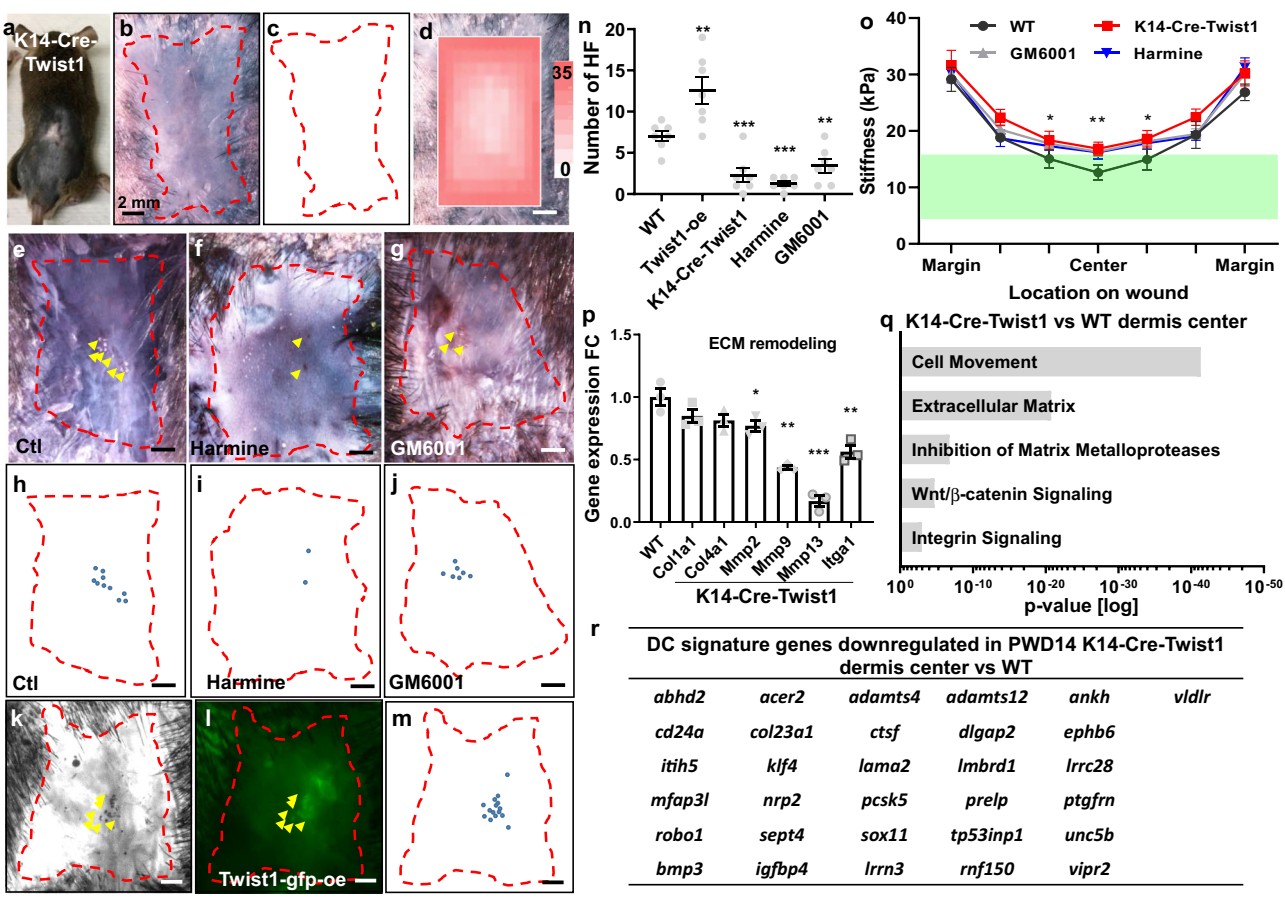

**Fig. 5 Epidermal Twist1 regulates both epidermal and dermal cell behavior and tissue stiffness towards hair primordia formation in WIHN.** **a** A K14-Cre-Twist1 mouse with a PWD28 wound. **b** AP staining of a K14-Cre-Twist1 PWD28 wound. **c**, Schematic diagram of **b**. **d** Stiffness heatmap of PWD14 K14-Cre-Twist1 wound overlaying on the wound photo. **e–g** AP staining of control (Ctl), and Harmine and GM6001 treated wounds on PWD28. **h–j** Schematic diagrams of **e–g**, respectively. **j** Photo of PWD28 wound transfected with Twist1-overexpressing lentivirus. **k** bright field and **l** fluorescence image of GFP-tagged-Twist1 overexpressing (oe) virus in the PWD28 wound, colocalizing with hair follicles. **m** Schematic diagram of **k** and **l**. **n** Dot plot showing the resultant hair follicle number in wild type, K14-Cre-Twist1, Harmine-treated, GM6001-treated and Twist1-overexpressing (oe) virus treated PWD28 wounds. $n = 7$ biologically independent animals. Data are presented as mean values ± SEM. All comparisons made to WT. Twist1-oe: $p = 0.0074$, K14-Cre-Twist1: $p = 0.0008$, Harmine: $p < 0.0001$, GM6001: 0.0044. Unpaired two-sided $t$-test. **o** Changes in wound stiffness upon different perturbations. K14-Cre-Twist1, GM6001 and Harmine treatments all significantly increased the stiffness of the wound center region, marked by asterisks. Green bar: 5–15 kPa morphogenetic range. $n = 5$ biologically independent animals. Data are presented as mean values ± SEM. All comparisons made to WT of respective location on wound. Center: K14-Cre-Twist1, $p = 0.0006$; GM6001, $p = 0.0006$; Harmine, $p = 0.0018$. Center-left: K14-Cre-Twist1, $p = 0.01115$; GM6001, $p = 0.0137$; Harmine, $p = 0.0214$. Center-right: K14-Cre-Twist1, $p = 0.0083$; GM6001, $p = 0.006$; Harmine, $p = 0.0159$. Unpaired two-sided $t$-test. **p** qPCR analysis: ECM remodeling related-gene expression fold change (FC) of wild type epidermis wound center vs K14-Cre-Twist1 wound center. $n = 3$ biologically independent samples. Data are presented as mean values ± SEM. All comparisons made to WT. $Mmp2$, $p = 0.0483$; $Mmp9$, $p = 0.0014$; $Mmp13$, $p = 0.0005$; $Itga1$, $p = 0.0071$. Unpaired two-sided $t$-test. **q** PWD14 wild type vs K14-Cre-Twist1 dermis wound center RNA-seq analysis identifies significantly enriched pathway affected by K14-Cre-Twist1 knockout. **r** Dermal condensate (DC) signature genes downregulated in PWD14 K14-Cre-Twist1 dermis wound center. Red dotted line: wound boarder. Yellow arrow: hair follicle. Harmine: Twist1 inhibitor. GM6001: pan-MMP inhibitor. Ctl: control. The images in **a–m** represent 6 out of 7 experiments performed.

To explore the signaling molecules perturbed by Twist1 knockdown during WIHN, we compared the gene expressions of PWD14 wild type epidermis wound center to that of K14-Cre-Twist1, and found downregulation in many genes related to cell proliferation, Wnt/β-catenin signaling (Supplementary Fig. S4f, g) and ECM remodeling (Fig. 5p). Previous studies have demonstrated that the epidermal placodes are required for underlying dermal condensation and the ensuing hair follicle development[41,42]. To further explore whether epidermal Twist1 affects DC fate acquisition during WIHN, we performed RNA-seq analysis on PWD14 wild type dermis wound center, PWD14 wild type dermis wound margin, and PWD14 K14-Cre-Twist1 dermis wound center. There

are 1623 DEG between PWD14 wild type dermis wound center vs margin (Supplementary Fig. S5a). Among them, DC signature genes are significantly enriched ($p = 1.06 \times 10^{-5}$), which suggests DC niche formation in the PWD14 dermis wound center. In addition, we found 928 DEG between K14-Cre-Twist1 and wild type dermis wound center comparison (Supplementary Fig. S5b), and the significantly enriched pathways include cell movement, ECM, inhibition of MMP, Wnt/β-catenin signaling and integrin signaling (Fig. 5q). To further investigate the contribution of epidermal Twist1 to dermal condensation, we overlapped DEG from wild type dermis center vs margin (total 1623 genes) and those from K14-Cre-Twist1 vs wild type dermis center genes (total 928

genes) (Supplementary Fig. S5c), and found 258 significantly overlapping DEG ($p = 1.37 \times 10^{-40}$). This indicates many DEG between wild type dermis wound center and margin were perturbed by epidermal Twist1 knockout. Among these DEG, most of the significantly upregulated genes were significantly downregulated by knocking out epidermal Twist1, and vice versa ($r = -0.61$, $p = 1.1 \times 10^{-27}$, Supplementary Fig. S5d). Similarly, we can identify 23 DC signature genes that are upregulated in the wild type dermis wound center vs wound margin, which were also downregulated by epidermal Twist1 knockout (Supplementary Fig. S5e). In total, there are 31 DC signature genes that are downregulated in PWD14 K14-Cre-Twist1 dermis center versus wild type (Fig. 5r). These results suggest that epidermal Twist1 plays an essential role in regulating DC and following hair follicle regeneration via an epidermal-dermal signaling interaction during WIHN.

Our findings here suggest tissue mechanics and epidermal Twist1 may feed in Wnt/β-catenin based hair primordia formation pathway in WIHN. This new "non-canonical" concept will be further discussed in discussion, together with literature.

**Turing-like mechanism explains an optimal wound stiffness range facilitates new hair placode formation.** From our previous mechanical analysis, we allude that an optimal range of wound stiffness is required for hair neogenesis to occur. This also suggests that in order for the hair placode to form and invaginate into the dermis, the aggregated cells should be able to overcome the physical barrier of the dermis. To quantify its mechanical properties, we used the AFM to map multiple $100 \times 100$ μm squares in the wound center to explore the apparent stiffness of the PWD14 hair placodes versus surrounding wound bed on a micrometer scale (Fig. 6a, left). The results show that the stiffness of hair placodes ($17.36 \pm 0.34$ kPa) is significantly higher than that of the wound center ($10.53 \pm 0.58$ kPa), but still much lower than the wound margin (Fig. 6a, dot plot), implying that the activated epithelial placode cells also undergo mechanical changes in addition to gene expressions (Fig. 6a′). In parallel, we also found that the average stiffness of E14 mouse embryonic skin to be $7.3 \pm 0.6$ kPa while its hair placode is $9.6 \pm 0.5$ kPa (Supplementary Fig. S6a, b), which is comparable to the microenvironment of the wound center in laboratory mice and the entire wound bed in spiny mice.

Turing model has been proposed as the underlying mechanism of pattern formation[43]. Here we look to construct a Turing system[44] to explain the differential placode formation pattern in laboratory and spiny mice, which is linked to the underlying structure of the solution region. Namely, the system should produce spots (hair placodes) within a specific region of stiffness; if the solution region is too soft, or too stiff, then the system does not pattern.

We consider three diffusible populations an activator, $u$, an inhibitor, $v$, and a measure of stiffness, $E$. Specifically, $u$, $v$ and $E$ are thought to be biochemical populations that are able to interact with each other. The prototypical "Schnakenberg" Turing kinetics[43,45,46] exist between the populations $u$ and $v$. The Schnakenberg kinetics are a general form of Turing kinetics, whereby all dynamics can be connected to source parameters, $\alpha$ and $\beta$. Since we have no guidance on kinetics these are as good as any kinetic type and swapping them for some more accurate kinetics should not influence the resulting conclusions.

We adapt the Schnakenberg kinetics by modulating the inhibitor source by the population $E$, which we take to be a stiffness measure of the field. Namely, a soft field has low density ECM and, thus, (we assume) the soft field produces more $E$, which, in turn produces more $v$. We consider a square spatial domain $[-50, 50] \times [-50, 50]$, centered at zero, with Neumann

boundary conditions and random initial conditions. In terms of the mathematics we produce the following system of interactions:

$$\underbrace{\frac{du}{dt}}_{\text{Rate of change of } u} = \underbrace{\nabla^2 u}_{\text{Diffusion of } u} + \underbrace{\alpha - u + u^2 v}_{\text{Schnakenberg interaction}} , \qquad (1)$$

$$\underbrace{\frac{dv}{dt}}_{\text{Rate of change of } v} = D_v \underbrace{\nabla^2 v}_{\text{Diffusion of } v} + \underbrace{E - u^2 v}_{\substack{\text{Schnakenberg interaction} \\ \text{with production} \\ \text{dependent on } E}} , \qquad (2)$$

$$\underbrace{\frac{dE}{dt}}_{\text{Rate of change of } E} = D_E \underbrace{\nabla^2 E}_{\text{Diffusion of } E} + \underbrace{sS(x,y)}_{\substack{\text{Region of} \\ \text{soft tissue} \\ \text{acts as a} \\ \text{source for } E}} - \underbrace{E}_{\substack{E \text{ decays over time}}} . \qquad (3)$$

The coefficients $D_v$ and $D_E$ are positive constants, which measure how quickly the populations spread. The source coefficient, $s$, measures the strength of the $E$ source,

$$S(x,y) = \exp\left(-\left(\frac{x}{\sigma}\right)^2 - \left(\frac{y}{\sigma}\right)^2\right). \qquad (4)$$

This means that the $E$ source is a scaled Gaussian distribution centered at zero. The source is highest in the center and decays towards the boundary. Physically, this means that the tissue is softest in the center and stiffest on the boundary. The 'size' of the soft region is controlled by $\sigma$, namely, increasing $\sigma$ makes the source function 'wider' meaning that more of the tissue is soft. An illustrative example of $S$ is shown in Fig. 6B.

Simulations of increasing $s$ can be seen in Fig. 6B. The top row (pink background) shows the output $u$ after a threshold (color change) has been applied, illustrates the change in tissue stiffness. The bottom row (blue background) shows the accompanying profile of $E$, reflects spot pattern formation. Namely, the spots in the blue background illustrate the regions in which $u > 3.8$. Once again, as $E$ decreases (from pink to green and blue) we expect the tissue beneath to be softer. Thus, we see that as $s$ increases left to right the center becomes lighter and lighter (pink background blocks). Notably, as the center becomes too soft (the last 2 pink background blocks in the third row) we see that placodes stop forming in this region (the last 2 blue background blocks in the bottom row). Hence, we have produced a simulation in which a Turing pattern has a feedback loop with an external field, which is considered to be a measure of the underlying ECM stiffness. Critically, in order for spots to form the media can be neither too stiff nor too soft.

**Discussion**

In summary, we show that multiscale tissue mechanics of the wound bed partake in setting up the morphogenetic field for WIHN, and Twist1 is an important chemo-mechanical regulator involved in initiating cellular events that lead to placode formation and invagination through symmetry breaking of the epidermis, ECM remodeling, collective migration and epidermal-dermal crosstalk. As we try to recapitulate the developmental process to facilitate regenerative wound healing, the mechanical microenvironment of the tissue should also be considered. The stiffening of the epithelial hair placode serves as an important symmetry breaking point - cells invaginating into the soft dermis (Fig. 6a). Identifying 5–15 kPa as the optimal stiffness range (Fig. 2k) is important in setting up the morphogenetic field for the hair placodes to overcome the physical barrier of the microenvironment and invaginate into the dermis. Furthermore, different spatiotemporal dynamics of wound stiffness in different species predicate the distribution of morphogenetic field and placode formation (Fig. 6c, d).

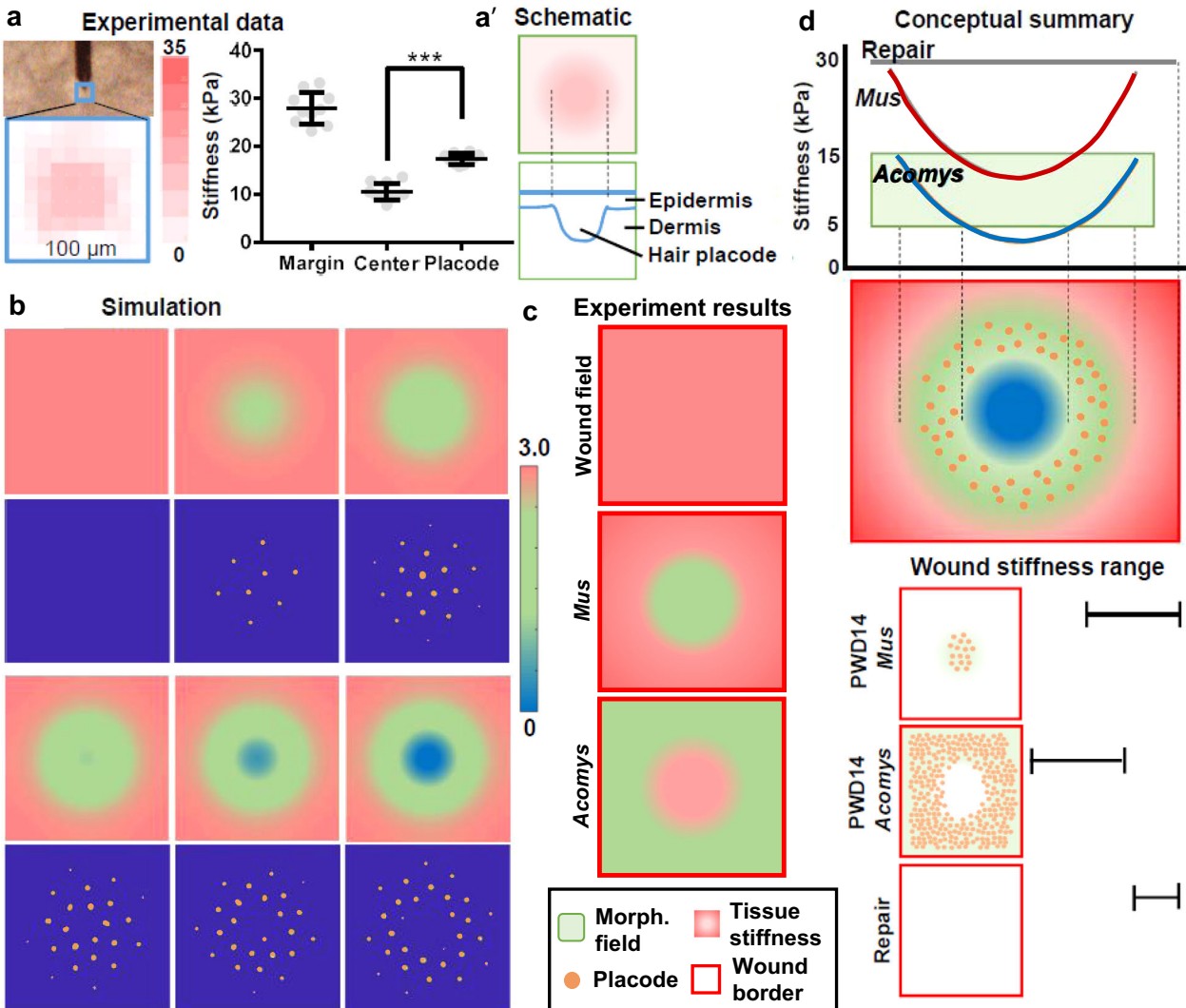

**Fig. 6 Multiscale tissue mechanics set up morphogenetic field for WIHN. a** AFM stiffness mapping demonstrates the stiffness of the hair placode. This is based on experimental data. Top left: an AFM cantilever scanning a PWD14 wound. Blue box demarcates the 100 × 100 μm scanning area. Below: stiffness heatmap of a placode. Colorimetric unit: kPa. Right panel: Dot plot showing the stiffness of placode with respect to wound margin and center. Representative image of 3 out of 3 experiments. n = 9 regions examined over 3 independent biological animals. Data are presented as mean values ± SD. p < 0.0001, unpaired two-sided t-test. **a′** Schematic diagram of placode stiffness and its respective cross-sectional view. **b** Hypothetical model showing feedback loop in a Turing system with an underlying measure of stiffness. Simulations illustrating stationary distributions of u (activator, blue background blocks) and E (stiffness, pick background blocks) with increasing s (source coefficient, measures the strength of the E source). The underlying domain is a square of side length 100 and the color shows the value of S at each grid point, S(x, y). The 'size' of the soft region is controlled by σ. From left to right and then the row below, the values of s are 2, 4, 6, 8 and 10, and σ = 20. In all cases random initial conditions from a uniform distribution of [0.5, 1.5] are used. **c** Schematic drawing showing the opposite topology of morphogenetic competent and non-competent region in the wound bed. **d** Conceptual summary of the way we perceive the relationship between tissue stiffness and morphogenetic field (Top). Summary based on data from Figs. 1 and 2. Middle: It highlights the different geographic distribution of the morphogenetic field (green) within a wound bed (red frame), and also the periodic appearance of hair primordia (orange) within the morphogenetic field (green). When the wound stiffness is too low (blue), no placode can form. Bottom: Stiffness of different wound beds predicate distribution of morphogenetic field and placode formation.

We demonstrate that there are two levels of symmetry breaking during successful WIHN, in parallel to the developmental process. The first level is the generation of morphogenesis competent field (green) from the center (Fig. 6b, c). The second level is the generation of periodically arranged hair germs forming (brown dots, Fig. 6b, blue blocks) from the morphogenesis competent field. In the spiny mice, the topology is reversed with the competence zone (green) on the periphery, while the central field (pink) cannot form hairs (Fig. 6c). By perturbing tissue stiffness, we can even generate a concentric ring-shaped competent field, fulfilling the prediction of the model (Fig. 6b, d). Additionally, the Turing mechanism can also help explain the asymmetric field in the less uniform environment (e.g., wound), development and growth[47].

The key question is what factors are required to make a region competent to undergo further periodic Turing patterning to generate hair placodes in the adult skin. The multiscale tissue mechanics perspective allows us to compare the similar and distinct pathways in development and WIHN, and appreciate that laboratory and spiny mice have evolved and manifested during regenerative wound healing, in contrast to repair. The findings and concept, together with those in recent WIHN studies are discussed in the following.

**Macroscale symmetry breaking of tissue mechanics in the wound bed leads to the emergence of morphogenetic field.** By comparing the mechanical and molecular response of the laboratory and spiny mouse during wound healing, we found that the hair placode formation pattern is opposite in laboratory and spiny mice during WIHN and this spatial difference is predicated by the wound bed with stiffness between 5–15 kPa, which is optimal for hair neogenesis and also the spatiotemporal expression of Twist1-related TFs.

We postulate that the spatiotemporal formation of the hair placodes is a good indication of the morphogenetic field of the wound bed, established partly by tissue mechanics. Previous studies have shown that there is an optimal matrix stiffness for different cell behaviors (e.g., cell migration, proliferation, differentiation), and very soft matrix impedes focal adhesion formation and cell migration[48]. We postulate that on PWD14 the wound center of the spiny mouse was still too soft for epidermal cells to form the hair placode, a process that requires epithelial cells to migrate and invaginate into the dermis, hence it occurred later on PWD21 when the wound stiffness reached over 5 kPa. On the other hand, a wound bed stiffer than 15 kPa may be too stiff for the epidermal hair placode cells to invaginate, as demonstrated by the thick and dense collagen fibers in the PWD14 laboratory mouse wound margin (Fig. 1l–s). The high collagen III expression in the spiny mouse wound in PWD14 and PWD21 (Fig. 2i), in contrast to the very few detectable collagen fibrils by SHG (Fig. 2h, Supplementary Fig. S2e, f), suggest that collagen III (not very crystallined and generates little SHG signal[38,39]) may be responsible for constituting the soft wound bed that resembles the physical environment of the embryonic skin, ideal for hair neogenesis. Recent studies also showed that ECM stiffness and mechanical forces exerted from the epidermal cells cohesively instruct tissue architectures and function[49,50].

Having a soft wound bed may also modulate the signaling of ECM remodeling gene expression. The time-course RNA-seq analysis of the spiny mouse wound showed an 1-week delay in the peak expression time of Twist1-related genes (PWD14) and ECM, MMP, integrins and other TFs (PWD21) in spiny mouse (Fig. 4b–d). We speculate that a suitable tissue stiffness is also required for TF to enter the nucleus, as supported by findings that force-induced nuclear deformation modulates nuclear entry of TF[51]. Hinz has also proposed a candy wrap theory to describe that certain level of mechanical force is required to release active TGF-β from its latent form[52]. The spiny mouse dermal cells also demonstrate fewer α-SMA-positive stress fibers upon substrate stiffness increase[53]. The spiny mouse's ability to keep the wound bed soft during early stages of wound healing could delay the nuclear entry and activation of the upstream TF and hence the expression of certain ECM, MMP and integrins, and consequentially set up the optimal molecular and mechanical wound bed as morphogenetic field. Furthermore, it is worth noting that Twist1 expression is higher in the dermis than epidermis (Fig. 4e) in spiny mouse around the hair placode, which is more representative of the embryonic dermal Twist1 expression during development (Supplementary Fig. S3d). These features could be the evolutionary advantages that the spiny mice have evolved to promote regenerative wound healing and survival. These similar yet distinct regulations of Twist1 and tissue mechanics between laboratory and the spiny mice remain to be investigated.

**Microscale symmetry breaking of tissue mechanics in the morphogenetic field leads to the emergence of hair primordial.** We have identified Twist1 as the key upstream chemo-mechanical regulator to activate ECM remodeling, epithelial cell movement, epithelial proliferation, Wnt/β-catenin signaling, and

DC in the dermis (Figs. 3–5). Analysis from a previous microarray database also showed that Twist1 expression is higher in the mice strain with high regenerative capacity compared to a low capacity strain (Supplementary Fig. 4e)[40]. Twist1 is shown to directly regulate *Cdh11*, *Grem1*, *Zeb1*, *Dkk3*, *Gli1*, *Fgfr1*, *Tbox18*, *Col6a2*, and *Lamb1* (Supplementary Fig. 7) when we reanalyzed a database that used H3K4me3 ChIP-seq to show the epigenetic reprogramming following Twist1-mediated EMT in human epithelial cells[54]. Others have also shown direct transcriptional binding of Twist1 on the Snai2 promoter[55]. This effect of Twist1 also corroborates with the detected increase in the stiffness of the placode. At the same time, Twist1 has also been shown to directly bind to the MMP promoter to exert its transcriptional effect[56]. By inducing MMP activity at the hair placodes and remodeling local ECM, we postulate this can lower the physical barrier of the dermis to also facilitate placode invagination. In this study, we used small molecule inhibitors Harmine and GM6001 to inhibit Twist1 and pan-MMP activities, respectively. Harmine targets the Twist1 pathways through its promotion of Twist1 protein degradation[57] and is also capable of blocking the activities of dual-specificity tyrosine phosphorylation-regulated kinase family proteins and mitogen activated protein kinase[58]. GM6001 is a potent reversible broad spectrum inhibitor of zinc-containing proteases, including various MMPs (MMP-1, −2, −7, −8, −9, −12, −13, −14, −16, and −26), disintegrin and metalloproteinase domain-containing (ADAM) proteins ADAM9, ADAM10, ADAM12, and ADAM17[59]. Although the results of the inhibitor treatment fell in line with our hypothesis and other Twist1-functional perturbation studies (lentivirus transfection and K14-Cre-Twist1 transgenic mice), and there is no significant difference between the number of new hair follicles observed in K14-Cre-Twist1 and Harmine treated wounds (Fig. 5n), we acknowledge these inhibitors' potential side effects outside of Twist1 pathway activities.

Furthermore, in order for the epithelial cells to continue to invaginate downward into the dermis, the cell number needs to increase, which can be observed in the highly enriched epidermal proliferation and movement genes in the gene set enrichment analysis (Fig. 3b). Similar expression of Twist1-related genes at the hair placodes have also been observed in our spiny mouse RNA-seq analysis (Fig. 4b), others' laboratory mice WIHN microarray database[60] and human hair follicle morphogenesis[40]. Our K14-Cre-Twist1 RNA-seq analyses also imply that epidermal Twist1 plays a role in dermal-epidermal interaction (Fig. 5q, r). The well-established morphogenesis initiator of skin development, β-catenin, has been shown to directly activate Twist1 expression in skull progenitor cells[61]. Therefore, we reason that Twist1 is one of the chemo-mechanical regulators that responds to β-catenin activation to induce symmetry breaking of the morphogenetic field of the epidermis, which facilitates dermal–epidermal interactions and initiates ECM remodeling, cell proliferation and collective migration, leading to placode formation and invagination during WIHN. Alternatively, Twist1 itself could also play the role of a mechanosensor during mechanotransduction for the wound induced hair follicle neogenesis[62–65].

**WIHN studies identify the concept of noncanonical and canonical hair primordia formation pathway.** WIHN is a combination of local periodic patterning events and a global influence that constitute the morphogenetic field. The canonical Wnt, β-catenin and Shh[66] have been identified as the critical activators of WIHN. Ablation of Wnt in the wound epidermis via inducible β-catenin deletion eliminates hair neogenesis, while overexpression of Wnt in the wound epidermis enhances it[22,67].

On the other hand, non-canonical signaling also regulates WIHN, some by interacting with the canonical signals. The δαT immune cells secret Fgf9 to act on the neighboring myofibroblasts in the wound, inducing them to secret Wnt2a ligand[60]. Double stranded RNA, which is released during injury, activates Toll-like receptor 3[40] and its downstream effectors IL-6[68] and Stat3 to promote hair neogenesis. This effect is achieved through the induction of known hair morphogenetic molecules such as Edar, Wnt, and Shh pathways.

Based on our earlier developmental studies of periodic formation of the feather and hair germs, we noticed there are two waves of molecular expression, which we name them restrictive and de novo mode, respectively[4]. In short, restrictive mode molecules are present before periodic patterning occurs, and are required for periodic patterning process, while de novo molecules are the readout. Our data showed Twist1 is expressed higher in the wound center (Fig. 3f), it is not exclusively expressed in the nucleus of hair placode cells, but also inter-follicular epidermis (Fig. 3f, g). These results suggest that Twist1 belongs to the "restrictive mode" molecules, and therefore they are present in the epidermis in both putative placode and inter-placode regions. Our view is that Twist1 is initially broadly expressed in the morphogenetic zone of the wound bed, and become accentuated in the placode region and enter the nucleus when the sum of all the upregulating factors for placode formation reaches a threshold. This is supported by our experiments where either softening the wound bed or overexpressing Twist1 in the wound enhanced WIHN. In spiny mice, we have noted that epidermal Twist1 is expressed predominantly but not exclusively in basal cells (Fig. 3f, g). While we postulate that activation of Twist1 in the basal cells suggests a prelude for EMT, recent discovery in the heterogeneity of wound epidermal cells[69] eludes that a more comprehensive and single-cell resolution future study is required to identify the molecular identities of these Twist1-positive and placode forming cells.

In this study, we show that Twist1 plays a key role in epidermal and dermal signaling during wound-induced hair primordia formation, which is a different mechanism adopted during developmental process. Studies have identified the binding sites for β-catenin on Twist1 promoter[61], although in our K14-Cre-Twist1 gene expression analysis, knocking out Twist1 also affected Wnt/β-catenin expression in both the epidermis and dermis during WIHN. We postulate β-catenin may be the initial activator of Twist1, and Twist1 can also loop back to regulate canonical Wnt/β-catenin signaling. These findings provide insights on the canonical/non-canonical molecular events during WIHN. Furthermore, we also demonstrate that tissue mechanics, or the stiffness of the wound bed, the ECM, and the stiffness of the cells also partake during WIHN; the hair primordia formation is not exclusive to molecular signaling events. The soft and easy-to-shed feature of the African spiny mouse skin not only serves as an escape strategy from predators[27], but also fosters the optimal mechanical cue for hair primordia formation during wound healing. Understanding the common and distinct features of laboratory and spiny mice in response to wounding shed light on evo-devo advantages and provide perspectives for future implications.

## Methods

**Animal model**. All animal work was performed according to the approved animal protocol, guidelines and regulations for the care and use of laboratory animals of University of Southern California (USC). Ethical approval was obtained for all experiments performed. All mice were housed in climate controlled indoor facilities in a temperature range between 21 and 26 °C with a 12:12-h controlled dark/light cycle. Humidity is maintained at 30–70%. C57BL/6J mouse purchased from Jackson Lab was used as the primary animal for this study. The K14-Cre-Twist1 mice were bred by crossing the Twist1 conditional null (Twist1$^{+/+}$)[70] and Tg

(KRT14-cre)1Amc/J (Jackson Lab) mice. The wild type Twist1$^{+/+}$ mice were used as the control in transgenic mouse study. The African spiny mouse, *Acomys cahirinus*, is a kind gift from Dr. Malcolm Maden at the University of Florida and Dr. Ashley W Seifert of University of Kentucky. A colony of captive-bred *Acomys cahirinus* was established at USC, and all experiments were per formed with protocols approved by the USC IACUC. We used both male and female 2-month-old spiny mice and 4-week-old C57Bl/6J mice for wound experiments unless otherwise specified.

**Wound-induced hair neogenesis assay**. Mice were anesthetized using Ketamine–Xylazine (80 mg/kg; 5 mg/kg) and analgesic Buprenorphine SR (0.5 mg/kg) was given by intraperitoneal injection (IP) at the beginning of the procedure. A 1 × 1 cm square full thickness wound was excised on the posterior dorsum of 4-week-old mice (p28) using scissors, and let it heal by secondary intention. For the spiny mice, 1.5 × 1.5 cm square full thickness wound was excised on 8 week-old mice. Additional DietGel Boost (ClearH2O) was placed on the bottom of the cage during the first week post-operation. Mice of the same sex from the same litter were housed together and provided with half-dome shelter.

**Alkaline phosphatase (ALP) stain**. To detect newly forming dermal papillae, alkaline phosphatase staining was performed as previously reported[22]. Briefly, full thickness wounds were excised and epidermis separated from the dermis using 20 mM EDTA. The dermis was fixed in acetone overnight at 4 °C, and washed in PBS several times. The dermis was pre-incubated in ALP buffer (0.1 M Tris-HCl, 0.1 M NaCl, 5 mM MgCl₂ and 0.1% Tween-20) for 30 min, incubated with BCIP/NBT Color Development Substrate (Promega, Madison, WI, USA) in ALP buffer at 37 °C until color development. The reaction was stopped by washing with pH 8.0 Tris-EDTA and the tissue stored in PBS with sodium azide.

**Atomic force microscopy**. AFM (NanoWizard 4a/CellHesion, JPK, Berlin, Germany) was setup for contact mode indentation in PBS. The spring constants of all cantilevers were calibrated via thermal noise method with correction factor in liquid[71,72] prior to each measurement resulting in values of 0.03 N/m. To allow for proper modeling of the data, a glass bead (5 μm in diameter) was attached at the end of a tipless rectangular cantilever (Arrow-TL1, NanoWorld, Neuchatel, Switzerland) using 2-component epoxy (Gorilla Glue Epoxy Clear, Gorilla Glue Company, Cincinnati, OH, USA). A force series identified a maximum indentation force of 5 nN to show the most consistent results on test samples. A constant rate of 1 μm/s was used for the entire approach and retract sequence. Force-distance curves were collected and post-processed using the JPK package software (Data Processing, 6.3.11). The force curves were analyzed using the Hertz model with a spherical indentation[35,36].

The force on the cantilever F($h$) is given by:

$$F(h) = \frac{E_{sample}}{1 - v_{sample}^2} \frac{4\sqrt{R}}{3} h^{3/2}$$

where $h$ is the depth of the indentation, $E$ is the effective modulus of a system tip-sample, $v$ is the Poisson ratio for the sample, and $R$ is the radius of the AFM tip. The unit of Young's modulus is calculated as N/m², and expressed as pascal (Pa) or kilopascal (kPa). Poisson ratio was set at 0.5 since the spherical tip was incompressible relative to the sample. The temperature of the measurement was controlled at 32 °C to mimic the surface temperate of mouse skin[73].

To maintain the biomechanical force integrity of the dorsal wound, the entire mouse skin organ was removed by creating an excision on the ventral side midline, extending from the anterior neck region to the posterior genital region, and then dissecting away the skin organ from the underlying fascia. Normal skin and wound stiffness were immediately measured after skin organ removal to prevent artifacts from tissue decomposition. AFM measurements were positioned across the wound starting from unwounded normal skin on one side and progressively traveling through to the opposite side; the near wound edge, the near wound bed, the wound center, the opposite wound bed, the opposite wound edge, and the opposite normal skin. At least 5 indentation points were taken for each region of interest.

**Second harmonic generation imaging**. The animals were anaesthetized using Isofluorane. Body temperature was maintained with a homeothermic blanket system (Harvard Apparatus, Holliston, MA, USA). The SHG images of animals were acquired using the external detectors of an inverted Leica SP5 (Wetzlar, Germany) multiphoton confocal fluorescence microscope powered by a Chameleon Ultra-II MP laser at 860 nm and a 40x Zeiss water-immersion objective (NA1.2). A Z-stack series of 3 μm per slice, 50 slices in total was recorded during a time course starting at the cornified layer of the epidermis and ending at 150 μm depth for each time point.

**Heatmap of spatial stiffness of the wound**. The interpolation of tissue stiffness was performed by using 3-D meshgrid function of MATLAB (R2015b, Natick, MA, USA). After obtaining a Young's modulus ($z$) at a specific spatial location ($x$, $y$) in the wound, a 3-dimensional matrix was defined. When the positions and stiffness of all the measured spots were identified, we could interpolate the stiffness of the

positions in between to average the stiffness of the nearest parameters using 3-D meshgrid function, by defining $(x, y)$ as meshgrid and $(z)$ as griddata. In the end, the heatmap was generated by defining the representative color of stiffness.

**Wound area calculation and stiffness analysis**. The area of the wound is quantified by using the area measurement function under ImageJ according to its user guide ImageJ/Fiji 1.46 (NIH, Bethesda, MD). The photo of the wound was taken with a scale bar, hence the measured area in pixel unit can be converted to actual size.

To obtain the area of the wound under 15 kPa, we specifically adjusted the $(x, y, z)$ values according to each respective wound so the heatmap was also actual size. Using Photoshop, we use the wand tool and set the tolerance value to match that of 15 kPa on the scale bar. By using this parameter, the wand tool could select the area of the wound along the 15 kPa line. This selected area is then saved and quantified using ImageJ to obtain the final actual area.

**Inhibitor treatments**. Blebbistatin (Cayman, MI, USA), Harmine (Cayman, MI, USA) and GM6001 (Cayman, MI, USA) were dissolved in DMSO. 20 μl was applied once a day directly to the wound surface starting at post-wound day 10 (PWD10) and continuing until PWD16.

**Histological preparations**. The wound tissues were fixed in 4% PFA and dehydrated in a graded alcohol series. The tissue was cleared in Xylene and embedded in paraffin wax. 6 μm sections were cut on a microtome. H&E sections were performed according to accepted protocol. Whole-mount tissues were fixed in 4% PFA and then stored at 4 °C in PBS with NaAzide.

**Paraffin section and whole-mount immunohistochemistry**. Fixed tissues were permeabilized with methanol and blocked with 3% $H_2O_2$ for 30 min, and then serum blocked for 1 h. The primary antibody was added and incubated over night at 4 °C with agitation. The tissue was washed with TBST (Tris Buffered Saline Tween 20) and the secondary antibody was added for 1 h at room temperature. The tissue was washed with TBST and if utilized, a tertiary antibody was added for 1 h at room temperature. The tissue was washed and color was developed using the AEC kit (Vector Laboratories, CA, USA) or fluorescence was visualized with a fluorescence microscope. The whole-mount samples were cleared in a series of Glycerol-PBS until 100% Glycerol for imaging. The Twist1 (ab50887), Collagen I (ab34710) and Collagen III (ab7778) antibodies are from Abcam (Cambridge, MA, USA), MMP9 (N2C1, GTX100458) is from GeneTex (Irvine, CA, USA), Snai1 (13099-1-AP), P-cadherin (13773-1-AP), E-cadherin (20874-1-AP) and Zeb2 (14026-1-AP) are from Proteintech (Rosemont, IL, USA). The dilution ratio for section IHC was 1:50, 1:400 for wholemount immunostaining.

**Harvesting wounds from laboratory mice for RNA extraction**. The wound was harvested and a 3 mm diameter hole-punch biopsy was taken from the geometric center of the wound, and the remaining wound tissue was considered the margin. The epidermis and dermis were separated manually under a dissecting microscope. The dissected dermis tissues were immediately placed in liquid nitrogen for 30 s. The frozen tissues were then disaggregated individually using a mortar and pestle, and then collected into a 1.5 ml microtube. The dissected epidermis was also collected into a 1.5 ml microtube.

**Harvesting wounds from spiny mice for RNA extraction**. The spiny skin (shaved) or wound was harvested and placed epidermal side down in a 3 cm culture dish filled with a thin layer of 0.25% Trypsin-EDTA (ThermoFisher, Waltham, MA, USA). The level of trypsin should be just enough to cover the epidermis but not submerging the tissue. The tissue is incubated at 4 °C for 6–12 h, rinsed in PBS and the dermis-epidermis were separated manually under a dissecting microscope. The epidermis was collected into a 1.5 ml microtube. The dissected dermis was placed in liquid nitrogen for 30 s and disaggregated using a mortar and pestle, and collected into a 1.5 ml microtube.

**RNA extraction and RNA-seq**. The RNA was extracted using the RNeasy Mini Kit (QIAGEN, Hilden, Germany). 1 μg of total RNA from each sample was used to construct an RNA-seq library using TruSeq RNA sample preparation v2 kit (Illumina, CA, USA). Sequencing (75 cycles single-end or paired-end reads) was performed by USC Molecular Genomics Core using a NextSeq 500 sequencer (Illumina, CA, USA).

**RNA-seq and microarray analysis**. The mouse mm10 reference genome, and RefSeq genome annotation downloaded from the UCSC Genome Browser on 5 June, 2019 were used for RNA-Seq analysis[74]. The alignment, quantification, normalization, and differentially expression analysis were performed by STAR 2.6.1d[75], htseq-count 0.6.0[76], TMM[77], and edgeR 3.26.8[78], respectively. $P$ value or False discovery rate < 0.05 was set as a threshold to identify differentially expressed genes (DEG). The hierarchical clustering, Venn diagram, volcano plot, gene expression profile, and scatter plot were carried out by Partek Genomics Suite

7.18.0723 (Partek Inc. MO, USA). The pathway enriched analysis based on Fisher's exact test, and upstream analysis were performed by Ingenuity Pathway Analysis (Content Version: 60467501, Build: ing_beryl, Date: 11-20-2020; IPA, QIAGEN Inc. CA, USA). An HTMH form Perl CGI program was performed for the statistical significance of the overlap between two groups of genes (http://nemates.org/MA/progs/overlap_stats.html). The gene set of DC signature gene and EMT core genes were build based on Supplementary Table S1 of[79] and upregulated genes in Supplementary Table S1 of[80]. Public microarray data under GEO database (GSE50418) were re-analyzed using Partek Genomics Suite 7.18.0723 (Partek Inc. MO, USA).

**Lentivirus production and transfection**. The Twist1 overexpressing vector genome plasmid was cloned by inserting Twist1 promoter into the lentiviral backbone: 5′LTR-cPPT-Ubq-eGFP-P2A-Twist1-WPRE-3′LTR. Twist1 promoter was amplified from mouse genomic DNA, and the plasmid backbone was purchased from Addgene (Watertown, MA, USA). The empty backbone without Twist1 promoter insertion was used as control.

293T cells (ATCC® CRL3216™) at 50–60% confluency were transfected with 10 μg vector genome plasmid, 10 μg of packaging construct ΔR8.2, and 2 μg envelope plasmid pCMV-VSVG using the calcium phosphate method. 10 mM sodium butyrate was added to fresh media 16 h post-transfection and removed after 8 h. Virus-containing media was collected at 36 h post-transfection, sterile filtered, and ultracentrifuged on a 20% sucrose cushion at 110,863 g and 4 °C for 1.5 h before storing at −20 °C for up to 30 days or −70 °C indefinitely.

The virus was applied to the wound on PWD10 to infect the tissue. The efficiency of transfection can be visualized by detecting the eGFP intensity under a fluorescent microscope, and later verified by frozen section IHC.

**RT-qPCR**. RNA extraction was done with Zymo Research Direct-zol RNA Kits. Reverse transcription was done using Superscript III First Strand Synthesis kit. The RNA and cDNA concentrations were measured with the NanoDrop 2000 spectrophotometer and normalized between samples. Primers used for qPCR are listed in Supplementary Table S2. The Ct values were measured using the Agilent Mx3000P qPCR system. The relative quantification was done by pyQPCR Version 0.9 software.

**Statistics**. The Kolmogorov–Smirnov tests were conducted to test normal distributed random samples. Two independent sample $T$-tests (two-tailed) were used for comparing unpaired sample groups. For some datasets not equally and normally distributed, Wilcoxon rank tests were conducted using IGOR Pro or MATLAB to evaluate the statistically significant difference between two samples. Chi-square test was conducted in MATLAB. The photographs are representative samples of at least four replicates. Stiffness measurements were reported as the averaged of at least four independent samples. Hair follicle counts are reported as the average from at least 6 samples. Each bar on qPCR graph represents average and SE of three independent samples. All data is presented as mean ± SD unless stated otherwise. Results from student $t$-tests (two-tailed) with $p < 0.05$ was considered significant. *, $p < 0.05$. **, $p < 0.01$. ***, $p < 0.005$.

**Reporting summary**. Further information on research design is available in the Nature Research Reporting Summary linked to this article.

## Data availability
The authors declare that all other data supporting the findings of this study are available within the article and its Supplementary Information files, or are available from the authors upon request.

Bulk RNA-seq data have been deposited in the NCBI Gene Expression Omnibus (GEO) database under accession code: GSE159939.

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

## Acknowledgements

This project is funded by NIGMS, NIH (5R01GM125322-02). H.I.C.H. is funded by The Featured Areas Research Center Program within the framework of the Higher Education Sprout Project by the Ministry of Education (MOE), Taiwan, The Excellent Research Center Program by the Ministry of Science and Technology (MOST 108-3017-F-006-002), Taiwan, and NIGMS, NIH. We thank the University of Southern California's Norris Medical Library Bioinformatics Service for assisting with sequencing data analysis. Some bioinformatics software and computing resources used in the study are funded by the USC Office of Research and the Norris Medical Library. S.T. is supported by NIH T90 grant #DE021982.

## Author contributions

H.I.H., S.P.W., B.V.H., Y.C. Liang, I.M.S., S.T., S.K., and T.X.J. performed the experiments. Y.C. Lai, A.S., H.X., A.D.P. analyzed the data. T.E.W. generated the model. M.H., M.J.T., J.P.P., D.E., T.X.J., and C.M.C. contributed study design and equipment support. H.I.H., Y.C. Lai, B.V.H., and C.M.C. wrote the manuscript.

## Competing interests

The authors declare no competing interests.
