## [Peer Review File · Nature Communications]

Reviewers' Comments:

Reviewer #1:

Remarks to the Author:

This manuscript addresses a potential mechanism for why hair follicles only regenerate at the center of wounds in the WIHN assay. The investigators study skin stiffness changes during wound healing and correlate hair follicle formation with a range of stiffness between 5 and 15KPa. They study both the C57Bl mouse and the African spiny mouse. They then use bulk and single-cell RNA-seq with cells isolated from different mouse models and attempt to show that epidermal Twist1 is a master regulator of epidermal-dermal interactions that controls hair follicle placode formation in the wound center. They do not link the two parts of the manuscript very well as there are no data showing an effect of Twist1 on stiffness for example.

Their findings, especially the part about the optimal stiffness for placode formation, as well as the neogenic hair formation pattern in the spiny mice (from wound margin to wound center) are novel, creative and exciting. These findings shed light on the overlooked role of mechanical properties of the wound matrix on hair regeneration. The second part of the manuscript delves into the role of Twist1 in WIHN. The authors barely attempt to link Twist1 to wound stiffness and the the conclusions are not fully supported by the data.

Overall, the authors do a nice job of correlating skin stiffness with hair follicle formation in C57Bl and the spiny mouse. They also provide some functional data as they manipulate stiffness with Blebbistatin, which inhibits myosin and decreases stiffness. The treated mice develop more hair follicles and have wounds that are less stiff compared to controls.

Some specific questions:

How are values of 15 kPa and 5 kPa determined as the upper and lower limit of stiffness? It is not clear how these cutoffs were obtained. Also, please provide relevant statistics in Figs 1R,S,T and Fig2J.

It is not clear why the authors picked PWD14 as the timepoint for study, since at this time hair follicles are already forming. Most of the differentially expressed genes likely reflect the presence of hair follicles rather than the earliest genes directing hair follicle formation. It is not clear why they picked Twist 1 to study when many other genes seem to have the same P value as Twist1 but with much higher fold change. Along those lines, in Fig 3 C the P-value of all the genes is not clear. They all seemed to be somewhere between 0 and 0.2. This kind of graph seems confusing.

The Twist 1 story seems compromised because of the wide expression of this gene throughout skin. Fig 6B suggests that Twist1 and Zeb1 are broadly expressed in almost all the cells (green cells) from the wound area, though very minimal Twist1 expression is seen in the wound epidermis and the placode. On the contrary, high intensity of Twist1 signals are shown in the dermis, likely in the dermal condensate cells. Is it not likely that the Twist inhibitor Harmine did not just affect the epidermal Twist1 but also the dermal condensate? The changes in hair follicle number by deleting or overexpressing Twist1 in the epidermis are not impressive enough to reach the broadly sweeping conclusions.

In order to link the stiffness story to the Twist1 story, it would be important to show tissue stiffness in the Harmine and GM6001 treated wound bed or in the Twist1 KO mice. What was the size of the area that was between 5 and 15 kpa in these treated or KO mice and did the size correlate with neogenic hair number?

Another useful experiment would be to mechanically influence wound contracture by splinting the wound. Was this done?

Please comment on the specificity of GM6001. Non-specific MMP inhibition would also have effects on WIHN that may be outside of Twist1 and its pathways. Similarly, what is the specificity of Harmine?

Does lentiviral transduction overexpress Twist1 in dermis, epidermis, or both?

Reviewer #2:

Remarks to the Author:

This is a remarkable paper concerning WIHN in the lab mouse and comparing it to spiny mouse hair follicle regeneration which encompasses a huge range of analyses from biomechanical analyses to pharmacological interventions to transgenic knockouts to viral overexpression to RNAseq to single cell RNAseq to mathematical modeling.

They find that the difference in behavior after skin wounding between these two species is due to stiffness of the wounds, examine the effects of pharmacologically altering stiffness, identify Twist1 as a central player in these biomechanical differences, look at downstream targets of Twist1 with RNA seq and single cell RNA seq, examine the effects of Twist1 knockouts and overexpression and develop a Turing model to predict wound behavior and the formation of hair follicles under these conditions of biomechanical stiffness.

It will have a major impact on the field as there have been very few reports of the biomechanical properties of cells and relating that to regenerative phenomena since Wong et al., 2011 brought this to the forefront in skin wound healing (incidentally there has been another biomechanical paper on the biomechanical properties of fibroblasts from the lab mouse and the spiny mouse with regard to substrate stiffness using traction force microscopy (Stewart et al., J. Biomech. 81, 149-154, 2018) and this seems highly relevant to this work, but it is not referred to).

Detailed comments

1. The writing is frequently ungrammatical throughout which constantly makes you stop and think what the authors are getting at and there are quite a few typos throughout the ms. My favourite is in line 45 (Abstract) which mentions 'a Turing model'. Turing would be turning in his grave knowing that his great contribution to developmental biology had been mis-spelt!

2. Line 118 – WIHN is highly dependent on the strain of laboratory mice used so I think it would be good to mention here at the start of the results which strain was used.

3. Lines 183 – 186 'there is a limit on the softness of the wound bed at 5 kPa in order for hair neogenesis to occur, as the initial hair placode can only be observed in wound stiffness between 5-15 kPa in both laboratory and spiny mice wounds (Fig. 2J)'. The blebbistatin experiment clearly work nicely for lab mice but for consistency does it work for spiny mice? Blebbistatin should decrease the number of hair placodes in spiny mice by taking the stiffness below the threshold for placode formation. Is this the case? This data would add weight to the stiffness argument. This data is also relevant to the Turing model and simulations developed in Fig. 7, so it is important for the whole thrust of the paper.

4. Lines 135-137 How specific is SHG for collagen 1? There have been several more recent analyses (than Gailitit & Clarke, 1994) of the presence of collagens in wounds, for example Brant et al., 2016 have found collagen 12 to be far more highly expressed in Mus wounds than collagen 1. So are you really only seeing collagen 1? I don't know the details of SHG so this is a question of ignorance on my part.

On the same subject - Line 179-180 – why are there no SHG images (Fig 2H) of spiny mice wounds as there are in the mouse wounds in fig 1N-Q? Are you suggesting there is no collagen 1 in spiny mice wounds? That is certainly not the case. Again, how specific is SHG for collagen 1?

5. Line 189 onwards, section on epidermal Twist1. This is a very difficult paragraph to follow. In the Mus RNA seq experiments first described, they were done on the epidermis and epidermal Twist1 is the 'second most significant upstream regulator'. In the spiny mouse RNA seq

experiments these were apparently done on both epidermis and dermis separately. In the epidermis Twist1 was again highly enriched in the epidermis (Fig. 4B) and in the dermis Twist1 was also enriched (Fig. 4C) – the title legend of Fig. 4C and 4D should include the word dermal in it to help the reader follow this data. Then IHC is done on a day 14 wound (Fig. 4E) and Twist is only seen in the dermis (top image in the column). Why is this when you have just emphasized the importance of Twist in the epidermis and dermis, I would expect to also see twist expression in the epidermis and it is not.

Then on lines 222-224 it states “It should be noted that while dermal Twist1 is important for hair formation in both developing placode formation and WIHN hair primordia formation, epidermal Twist1 is only seen in WIHN placode formation”. There is no data to support this as the RNA seq data from WIHN in Mus was only done on the epidermis with no mention of the dermal expression of genes, so how can you conclude that it is ONLY seen in the epidermis in WIHN?

6. Fig. 4E immunos. I don't find the difference in expression between E-cad ('downregulated at the hair placode') and P-cad ('expressed in and around the hair placode') to be convincing at all, they both look the same to me.

7. Lines 294-295, immunos in Fig 6G – J. Now twist is in the epithelial placode and not the dermis in contrast to the same immuno in Fig 4E. MMP9 (Fig. 6I) and P-cad (Fig. 6H) are also different from Fig 4E.

8. Lines 337-345. There isn't a Figure 8 so what does this refer to?

9. Simulations in Fig 7. Is the simulation for spiny mouse in the lower half of Fig. 7C with a ring of hair follicles surrounding a center with no hair follicles ever seen? The only comparable image of a spiny mouse wound with hair follicles in this manuscript is Fig. 2C and 4F. In both of these cases the hair follicles are throughout the wound bed and not absent from the center. So the real situation does not mimic the model.

Reviewer #3:

Remarks to the Author:

The manuscript by Harn et al. describes an investigation of the role of mechanochemical signalling in the regulation of wound healing.

Among other tools, the authors use scRNA-seq to study changes in cell populations during wound-induced hair follicle regeneration. Regrettably, these analyses are rather superficial and several key details are missing:

Figure 6A and related text:

The name of the dimensionality reduction method is not given. Neither is the name of the clustering method. How were the cell types identified? No data related to this is presented. In addition, some of the cluster names are not very informative, in particular “Twist1high”. Without such details it is impossible to meaningfully assess the validity of the conclusions.

Figure 6B:

The authors state that the genes presented in this panel are expressed in the transitional cell population. In my opinion the figure does not convincingly support this statement. At least, they seem to be expressed at equal or higher levels in several other cell clusters. In addition, two of the genes (Vcan and Crabbp1) are not discussed in the text and their significance is unclear. The cluster-related differences would be easier to interpret from e.g. a violin plot.

Figure 6D-J:

In my opinion, the relation of these results and those of the scRNA-seq analysis is questionable, as

the expression of Twist1 and Snai1 at RNA-level does not seem to be limited to any single cluster. Therefore, the connection of the cells studied by IHC and those analysed by scRNA-seq remains rather vague.

Figure 6 and S. Fig 5:

The lack of overlap between control and PWD14 fibroblasts is striking, and warrants more thorough analysis. The authors mention that Grem1 and Inhba are differentially expressed, but the expression of these genes (or other top hits) is not shown at single-cell level. Are they ubiquitously expressed or upregulated in a specific sub-population? Furthermore, it would be important to account for the role of technical effects: were the samples processed in the same batch or at different times? Is the technical quality similar?

Methods section:

- How many cells were sequenced from each condition?
- How deeply were the cells sequenced? How many genes were detected per cell?
- How was the data normalized? What units of gene expression are presented in 6B-C?
- Trajectory analysis is mentioned but it is not shown in the results section

Reviewers' comments:

We thank all the reviewers for their comments. Our responses are written in blue. Modified figures are marked by yellow highlights. Cited revised text is written in italic.

Reviewer #1

This manuscript addresses a potential mechanism for why hair follicles only regenerate at the center of wounds in the WIHN assay. The investigators study skin stiffness changes during wound healing and correlate hair follicle formation with a range of stiffness between 5 and 15KPa. They study both the C57Bl mouse and the African spiny mouse. They then use bulk and single-cell RNA-seq with cells isolated from different mouse models and attempt to show that epidermal Twist1 is a master regulator of epidermal-dermal interactions that controls hair follicle placode formation in the wound center. They do not link the two parts of the manuscript very well as there are no data showing an effect of Twist1 on stiffness for example.

Their findings, especially the part about the optimal stiffness for placode formation, as well as the neogenic hair formation pattern in the spiny mice (from wound margin to wound center) are novel, creative and exciting. These findings shed light on the overlooked role of mechanical properties of the wound matrix on hair regeneration. The second part of the manuscript delves into the role of Twist1 in WIHN. The authors barely attempt to link Twist1 to wound stiffness and the conclusions are not fully supported by the data.

Response: We appreciate the constructive comment. This issue is addressed by determining and connecting the effect of Twist1 on wound stiffness. RNA-seq analyses also suggest Twist1-dependent epithelial-mesenchymal transition may be involved in the non-canonical hair primordia formation as we observed here. More detailed responses are described in the “specific question” section below.

Overall, the authors do a nice job of correlating skin stiffness with hair follicle formation in C57Bl and the spiny mouse. They also provide some functional data as they manipulate stiffness with Blebbistatin, which inhibits myosin and decreases stiffness. The treated mice develop more hair follicles and have wounds that are less stiff compared to controls.

Some specific questions:

1. How are values of 15 kPa and 5 kPa determined as the upper and lower limit of stiffness? It is not clear how these cutoffs were obtained. Also, please provide relevant statistics in Figs 1R, S, T and Fig2J.

Range of kPa. The kPa numbers are derived from the stiffest (upper) and softest (lower) threshold of hair neogenesis in *mus* and *acomys*, respectively. We observed that there was no hair neogenesis in areas of the wound that are stiffer than 15 kPa, nor softer than 5 kPa (Fig. 2K). We can expand the wound area having a stiffness range between 5-15 kPa in *mus* by treating the wound bed with Blebbistatin and hair neogenesis is increased. On the other hand, Blebbistatin treatment in the *acomys* softened the wound bed stiffness to below 5 kPa and reduced hair neogenesis (Fig. 2J). Thus, either too stiff or too soft is not favorable for new hair formation, while the optimal range (5 -15 kPa) is just right. These data are now added to Fig. 2J, 2K, SI Fig. 2A-D and described clearer in the results section.

Statistics. The relevant statistics have been analyzed and now are included in Fig. 1R, S, T and 2J.

2. It is not clear why the authors picked PWD14 as the timepoint for study, since at this time the hair follicles are already forming. Most of the differentially expressed genes likely reflect the presence of hair follicles rather than the earliest genes directing hair follicle formation. It is not clear why they picked Twist 1 to study when many other genes seem to have the same P value as Twist1 but with much higher fold change. Along those lines, in Fig 3 C the P-value of all the genes is not clear. They all seemed to be somewhere between 0 and 0.2. This kind of graph seems confusing.

Choice of time point. Our initial time course study includes post wound day 10, 12 and 14. Morphologically, formation of the placode is more distinct on PWD14, but this time precedes further cell behavior and cell fate changes (SI Fig. 3B). The genes of interest to this study peak on PWD14 (SI Fig. 3A). It is not practical to do detailed studies at every time point, thus we picked PWD14 for further study here. To gain more understanding, we hope to extend the study to different time points in future studies. For now, we added this explanation to the results section and SI Fig. 3A-B of the manuscript.

Choice of Twist1: By comparing the bulk-RNA-seq analysis of the PWD14 epidermal wound at the center versus wound margin, we chose Twist1 for further study because it was identified by IPA (Ingenuity Pathway Analysis) as the second most significant upstream transcription factor (Fig. 3D). Indeed, there are many molecular pathways which might be involved in WIHN and are under current investigation by others and by our group. Twist1 is known to also function in dermal cells. Twist1-dependent epithelial-mesenchymal transition may be involved in the non-canonical hair primordia formation observed here.

Upstream regulator	p-value
NFKBIA	3.28E-13
TWIST1	4.5E-11
CEBPA	4.97E-11
SMAD7	3.44E-09
SNAI1	3.36E-08
HDAC1	5.78E-08
IRF7	2.09E-07
JUNB	6.87E-07
RELA	1.68E-06
SMAD3	2.31E-06

3. The Twist 1 story seems compromised because of the wide expression of this gene

throughout skin. Fig 6B suggests that Twist1 and Zeb1 are broadly expressed in almost all the cells (green cells) from the wound area, though very minimal Twist1 expression is seen in the wound epidermis and the placode. On the contrary, high intensity of Twist1 signals are shown in the dermis, likely in the dermal condensate cells. Is it not likely that the Twist inhibitor Harmine did not just affect the epidermal Twist1 but also the dermal condensate? The changes in hair follicle number by deleting or overexpressing Twist1 in the epidermis are not impressive enough to reach the broadly sweeping conclusions.

One of the key findings of this study is the small but important population of Twist1-expressing epidermal cells in the wound center that we proposed will form the hair placode. Without this activation in the epidermal cells, hair placodes cannot form (in the case of K14-Cre-Twist1 mice) and no hair neogenesis occurs.

Harmine can affect both epidermal and dermal Twist, and it is likely that dermal Twist may also play an important role in hair neogenesis. However, we have not observed dermal condensation in the numerous PWD14 histology sections despite the positive expression of Twist1 in the dermis. This implies that Twist1 is expressed in the epidermal cells before dermal condensation is formed in our model. Furthermore, bulk-RNA-seq analysis showed that knocking out Twist1 specifically in the epidermis with K14-Cre led to downregulation of DC signature genes in the dermis of PWD14 wound center (Fig. 5R). These results favor the explanation that epidermal Twist1 expression, despite its limited number of expressing cells, is essential for epidermal hair placode formation and also for dermal condensate formation in the dermis.

To strengthen the statistical power of the experiment, we performed additional rounds of experiments to increase the sample size of K14-Cre-Twist1, Harmine, GM6001 (an MMP inhibitor) and Twist1-overexpression groups. The updated results showed significant differences between the Twist1-perturbed groups and control. It is also worth noting that there is no significant difference between K14-Cre-Twist1 (epidermal Twist1 knockout only) and Harmine (both epidermal and dermal Twist1 inhibition) treated groups. This updated result is used to replace the previous Fig. 5N.

4. In order to link the stiffness story to the Twist1 story, it would be important to show tissue stiffness in the Harmine and GM6001 treated wound bed or in the Twist1 KO mice. What was the size of the area that was between 5 and 15 kPa in these treated or KO mice and did the size correlate with neogenic hair number?

As suggested, we have performed these new experiments accordingly. The tissue stiffness of the wound center in K14-Cre-Twist1, Harmine, and GM6001 treated mice were still relatively softer than the wound margin; however, the size of the area under 15 kPa is significantly smaller than control (Fig. 5O, SI Fig. 4B-C). There is a correlation ($R^2 = 0.7496$) between the wound area under 15 kPa and the number of HFs in the Twist1-perturbed wounds; however, the slope of the trend line is different from that of the control and Blebbistatin treated samples (1.68 vs 12.8, SI Fig. 4D, Fig. 1K).

SI Fig. 4D

Fig. 1K

5. Another useful experiment would be to mechanically influence wound contracture by splinting the wound. Was this done?

Thank you for this good suggestion. Splinting the wound is indeed something we also considered and tried. Unfortunately, the weight of the metal splint device is rather heavy for these 4-week old mice, and the weight of the device itself was causing the whole dorsal skin to stretch. Furthermore, splinting the wound in these mice causes physical damage to the skin and made the results uninterpretable.

6. Please comment on the specificity of GM6001. Non-specific MMP inhibition would also have effects on WIHN that may be outside of Twist1 and its pathways. Similarly, what is the specificity of Harmine?

GM6001. According to literature, GM6001 is a potent reversible broad spectrum inhibitor of zinc-containing proteases, including matrix metalloproteinases (MMPs) (Auge, Hornebeck et al., Bioorganic & Medicinal Chemistry Letters, 2003). GM6001 inhibits MMP-1, -2, -7, -8, -9, -12, -13, -14, -16, and -26. It also inhibits disintegrin and metalloproteinase domain-containing (ADAM) proteins ADAM9, ADAM10, ADAM12, and ADAM17. We acknowledge the possible side effects outside of the Twist1 pathway, and have included this concern in the discussion (*Paragraph 5, line 15-23*).

Harmine. Harmine targeted the TWIST1 pathway through the promotion of TWIST1 protein degradation (Yochum, Cades, et al. Molecular Cancer Research, 2017). Harmine is also capable of blocking the activities of dual-specificity tyrosine phosphorylation-regulated kinase (DYRK) family proteins and mitogen activated protein kinase (Uhl, Schultz, et al., Cancer Cell Int. 2018). While DYRK isn't known to be involved in WIHN, it is worth mentioning. We have included this in the discussion (*Paragraph 5, line 15-27*).

7. Does lentiviral transduction overexpress Twist1 in dermis, epidermis, or both?

Previously when we injected Lentivirus with the FUGW-backbone and stained the wound section with GFP, we are able to see the virus transfection mainly in the epidermis, though there are also a few GFP+ cells in the dermis (Fig. below). From our scRNA-seq and IHC data, we know that Twist1 is highly expressed in the wound dermis, hence we presume that most of the effects caused by the lentiviral Twist1 overexpression is from the epidermis. However, the concern is valid and is included in the materials and methods section (under the section about lentivirus production and transfection).

Reviewer #2

This is a remarkable paper concerning WIHN in the lab mouse and comparing it to spiny mouse hair follicle regeneration which encompasses a huge range of analyses from biomechanical analyses to pharmacological interventions to transgenic knockouts to viral overexpression to RNAseq to single cell RNAseq to mathematical modeling.

They find that the difference in behavior after skin wounding between these two species is due to stiffness of the wounds, examine the effects of pharmacologically altering stiffness, identify Twist1 as a central player in these biomechanical differences, look at downstream targets of Twist1 with RNA seq and single cell RNA seq, examine the effects of Twist1 knockouts and overexpression and develop a Turing model to predict wound behavior and the formation of hair follicles under these conditions of biomechanical stiffness.

It will have a major impact on the field as there have been very few reports of the biomechanical properties of cells and relating that to regenerative phenomena since Wong et al., 2011 brought this to the forefront in skin wound healing (incidentally there has been another biomechanical paper on the biomechanical properties of fibroblasts from the lab mouse and the spiny mouse with regard to substrate stiffness using traction force microscopy (Stewart et al., J. Biomech. 81, 149-154, 2018) and this seems highly relevant to this work, but it is not referred to).

We appreciate the constructive comments. This reference is now added to the discussion (4th paragraph, line 7-9).

Detailed comments:

1. The writing is frequently ungrammatical throughout which constantly makes you stop and think what the authors are getting at and there are quite a few typos throughout the ms. My favorite is in line 45 (Abstract) which mentions 'a Turing model'. Turing would be turning in his grave knowing that his great contribution to developmental biology had been mis-spelt!

We apologize for the typo. We tried to revise and polish the manuscript.

2. Line 118 – WIHN is highly dependent on the strain of laboratory mice used so I think it would be good to mention here at the start of the results which strain was used.

Yes, the results are affected by strain and age of the mice used. For this study C57Bl/6 strain is used. We now mention this as follows:

“we first created 1 x 1 cm full thickness wounds on the dorsal skin of 3-week-old C57Bl/6 mice”

3. Lines 183 – 186 ‘there is a limit on the softness of the wound bed at 5 kPa in order for hair neogenesis to occur, as the initial hair placode can only be observed in wound stiffness between 5-15 kPa in both laboratory and spiny mouse wounds (Fig. 2J)’. The blebbistatin experiment clearly work nicely for lab mice but for consistency does it work for spiny mice? Blebbistatin should decrease the number of hair placodes in spiny mice by taking the stiffness below the threshold for placode formation. Is this the case? This data would add weight to the stiffness argument. This data is also relevant to the Turing model and simulations developed in Fig. 7, so it is important for the whole thrust of the paper.

Thank you for the constructive comment. As suggested, we treated the spiny mice with Blebbistatin from PWD10-PWD17 and observed a significant decrease in regenerated numbers from 74 to 12.4 (Fig. 2J). We also measured the stiffness of the Blebbistatin-treated spiny mouse wound on PWD14, in which the area under 5 kPa increased significantly (SI Fig. 2A-D).

The result suggests the optimal range of wound stiffness for hair neogenesis is between 5 -15 kPa. Regions of wounds that are too rigid or too soft are not favorable for new hair formation.

4. Lines 135-137 How specific is SHG for collagen 1? There have been several more recent analyses (than Gailitit & Clarke, 1994) of the presence of collagens in wounds, for example Brant et al., 2016 have found collagen 12 to be far more highly expressed in Mus wounds than collagen 1. So are you really only seeing collagen 1? I don't know the details of SHG so this is a question of ignorance on my part. On the same subject - Line 179-180 – why are there no SHG images (Fig 2H) of spiny mice wounds as there are in the mouse wounds in fig 1N-Q? Are you suggesting there is no collagen 1 in spiny mice wounds? That is certainly not the case. Again, how specific is SHG for collagen 1?

Thank you for the opportunity to clarify. SHG is not specific for collagen I, but it works best for fibril collagens (type I, II, III and V). Type III and V collagen are less crystalline and generate little or no SHG signal, and type II collagen is abundant in cartilage and not prominent in the skin. We cannot completely rule out that SHG imaging could also pick up collagen types other than collagen I. We now include the following sentences in the results section: “Collagen has been implicated as the main extracellular matrix (ECM) of the wound, hence we used second harmonic generation (SHG) to visualize the amount and organization of the collagen fibrils in the wound (Fig. 1L-Q).”

For the wound in the spiny mice, our IHC staining against collagen I and III showed there is some, but not a significant amount, of collagen 1 in post-wound week 2 and 3 wounds (Fig. 2I). We suspect this low level of collagen 1 could be insufficient for the SHG to detect and form an image. We revised the description in Line 176 to 182 to: “To investigate the structure and

organization of fibril collagen in the spiny mouse wound as observed in laboratory mice, we also used SHG to visualize it. Interestingly, very little SHG signal was detected within the wound bed in both PWD14 and PWD21 spiny mouse wounds (Fig. 2H). Spiny mice skin has been reported to have high levels of collagen III²⁰; however, since type III are less crystalline and generate little SHG signal^{31,32}, we used IHC to examine the expression of collagen I and III in the spiny wounds. “

5. Line 189 onwards, section on epidermal Twist1. This is a very difficult paragraph to follow. In the Mus RNA seq experiments first described, they were done on the epidermis and epidermal Twist1 is the ‘second most significant upstream regulator’. In the spiny mouse RNA seq experiments these were apparently done on both epidermis and dermis separately. In the epidermis Twist1 was again highly enriched in the epidermis (Fig. 4B) and in the dermis Twist1 was also enriched (Fig. 4C) – the title legend of Fig. 4C and 4D should include the word dermal in it to help the reader follow this data.

Thank you for the opportunity to clarify. The titles of Fig. 4B, 4C and 4D have been modified to the following:

Fig. 4B: Proliferation of epithelial cells and cell movement associated genes in the spiny mouse wound epidermis

Fig. 4C: Twist1 related transcription factors in the spiny mouse wound dermis

Fig. 4D: ECM remodeling in the spiny mouse wound dermis

Then IHC is done on a day 14 wound (Fig. 4E) and Twist is only seen in the dermis (top image in the column). Why is this when you have just emphasized the importance of Twist in the epidermis and dermis, I would expect to also see twist expression in the epidermis and it is not.

Thank you for pointing this out. We used 2 different Twist1 antibodies and performed IHC on PWD14 spiny mouse wounds. As seen below, although Twist1 is highly expressed in the dermis, there is still some Twist1-expressing cells in the epidermis, especially in the epithelial placode (see below).

Then on lines 222-224 it states “It should be noted that while dermal Twist1 is important for hair formation in both developing placode formation and WIHN hair primordia formation, epidermal Twist1 is only seen in WIHN placode formation”. There is no data to support this as the RNA seq data from WIHN in Mus was only done on the epidermis with no mention of the dermal expression of genes, so how can you conclude that it is ONLY seen in the epidermis in WIHN? We apologize for the misleading description. What we meant to convey is that during normal skin development, Twist1 is only expressed in the dermis (SI Fig. 3C, below). During WIHN, Twist 1 appear in the epidermis in the novel non-canonical hair primordia formation we reported in this paper. We revised this as follows: *“It should be noted that while dermal Twist1 is highly expressed in embryonic skin, it is absent in the E14 epidermal hair placode (SI Fig. 3C). Identifying epidermal Twist1 in WIHN suggests that the spatiotemporal dynamics of tissue stiffness and gene expression may play a more important role in epithelial placode formation in WIHN than placode formation in embryonic development.”*

6. Fig. 4E immunos. I don't find the difference in expression between E-cad ('downregulated at the hair placode') and P-cad ('expressed in and around the hair placode') to be convincing at all, they both look the same to me.

Thank you for the opportunity to clarify. We have replaced the p-cad figure with another more distinct figure that shows p-cad expression at the hair placode and parts of the basal epidermal cells. We also use arrows to point out the downregulation of E-cad at the hair placode. These figures have been updated in new Fig. 4E.

Furthermore, we looked into the literature of E-cad expression in the developing hair placode. In Jamora et al. (Nature, 2003, Figure 2a,b), E-cad is also downregulated and P-cad upregulated at the epithelial hair placode.

[Image Redacted]

See:

Links between signal transduction, transcription and adhesion in epithelial bud development.
Jamora C, DasGupta R, Kocieniewski P, Fuchs E. Nature. 2003 Mar 20;422(6929):317-22. doi: 10.1038/nature01458. PMID: 12646922

7. Lines 294-295, immunos in Fig 6G – J. Now twist is in the epithelial placode and not the dermis in contrast to the same 11immune in Fig 4E. MMP9 (Fig. 6I) and P-cad (Fig. 6H) are also different from Fig 4E.

Thank you for pointing out the discrepancy. We have performed another round of IHC staining using newly purchased antibodies, and showed that both Twist1 and MMP9 are expressed in the hair placode and also in the dermis as well (Fig. 3F, 4E), which correspond to our data. Lastly, the difference in P-cad expression pattern between Fig. 4E and Fig. 6H could be due to the species difference in C57Bl/7 and spiny mice. Nevertheless, we have revised our Fig. 6 according to Rev 3's comments, and the previous data were removed.

8. Lines 337-345. There isn't a Figure 8 so what does this refer to?
Sorry, it was a typo. It's referring to Fig. 7.

9. Simulations in Fig 7. Is the simulation for spiny mouse in the lower half of Fig. 7C with a ring of hair follicles surrounding a center with no hair follicles ever seen? The only comparable image of a spiny mouse wound with hair follicles in this manuscript is Fig. 2C and 4F. In both of these cases the hair follicles are throughout the wound bed and not absent from the center. So the real situation does not mimic the model.

It is true that eventually the entire wound bed of the spiny mouse would regrow hair; however, the sequence of its hair placode formation begins from the wound periphery (PWD14) and then later in the wound center (PWD21). We would point out more clearly that our simulation model is addressing the formation of the morphogenetic field with respect to spatial tissue stiffness at a particular time point during wound healing, which is at PWD14. And the orange dots represent epithelial hair placodes and not the eventual hair formation.

Reviewer #3

The manuscript by Harn et al. describes an investigation of the role of mechanochemical signalling in the regulation of wound healing.

1. Among other tools, the authors use scRNA-seq to study changes in cell populations during wound-induced hair follicle regeneration. Regrettably, these analyses are rather superficial and several key details are missing:

We appreciate the reviewer's comments on the scRNA-seq data. The major point of this paper is the role of tissue mechanics in WIHN, and we have used different mouse types and gene-misexpression studies to demonstrate Twist1 is involved. Our motivation to do scRNA analyses is to gain insights on its mechanism but we have to acknowledge our scRNA analyses is preliminary and not comprehensive. The reviewer pointed out several issues with the analysis, including how clusters were labeled/identified and how these results related to the rest of the paper. Based on the reviewer's comments, we now revised our use and presentation of single cell data to better connect with the rest of the manuscript, namely to help determine how Twist1 in epidermis supports WIHN. We now use our analysis of naive, unwounded skin to identify genes enriched in DP cells (Fig. 6A). Then we compare it with a dataset published by Lim et al. which exhibits enhanced WIHN based on activated Shh signaling in SM22 (TAGLN)-expressing cells (Fig. 6B, SI Fig. 6A). In their data, we can identify a similar cellular cluster annotated in their paper as a cluster of DP-like cells (Fig. 6C-E, SI Fig. 6C marked by the expression of Prlr, Alpl, Wif1, Rspo3). We also identify a cell cluster almost exclusively present in SM22-SmoM2 wounds that has some epithelial character (Cldn1, CD200). Further analysis of the latter revealed significant gene expression overlap between these cells and the genes enriched in the center of the big WIHN wound, and differentially expressed genes between wild type vs. Twist1 KO mouse epidermis (Fig. 6F, SI Fig. 6D). We partially verified these results by IHC in WT and K14-Cre-Twist1 PWD14 wounds (Fig. 6G). Together, these data imply that during WIHN, epithelial cells acquire some mesenchymal gene expression and that this change requires Twist1, potentially via a partial EMT.

While these scRNA data are intriguing, we agree it is not rigorously proven because we do not yet have data using the Cre-ROSA26 reporter for direct lineage tracing which will take too long to acquire. The major point of this paper is that "symmetry breaking in tissue mechanics" promotes WIHN. We think we have provided sufficient evidence to support this statement. Then, we explored the mechanism of this activity. We are also confident in stating that Twist 1 is

involved. Trying to understand how it is involved was the original motivation for us to apply the newly available scRNA analyses. The emergence of new cell clusters based on our and Ito's scRNA analyses is indeed inspiring and can stimulate further research to characterize these "non-canonical" hair primordia formation pathways (please see discussion). It is not proven and we have indicated so in the text. We also have considered the possibility to remove Fig. 6 on scRNA analyses or put some panels in the supplement, or leave scRNA work for a future paper. Indeed, we are in the process of a study with scRNA, scATAC analyses to study WIHN with and without Blebbistatin, which can suppress non-muscle myosin and modulate tissue tension. We can do a more comprehensive scRNA analyses in that future paper. We think the basic theme and title of this manuscript can still stand without it. Yet, keeping the preliminary scRNA analysis in the current manuscript can make the work more interesting with potential mechanisms. We will appreciate your and editor's advice.

In addition to the added points above, we also provide a better description of scRNA data generation and analysis.

2. Figure 6A and related text: The name of the dimensionality reduction method is not given. Neither is the name of the clustering method. How were the cell types identified? No data related to this is presented.

We have added these parameters to the materials and methods. PCA was used to reduce dimensionality and t-SNE was used to cluster the cells. In the new analysis and presentation, the cell types were identified according to the markers expressed by the cell clusters. For instance, *Cdh5* and *Pecam1* were used to identify endothelial cells, *Myh11* and *Rgs5* for muscle cells, *Mbp* and *Plp1* for Schwann cells, and *Dpt* and *Pdgfra* for fibroblasts (SI Fig. 6B). *Prlr*, *Wif1*, *Enpp2*, *Hhip*, and *Dio2* were used to identify DP cells (Fig. 6A) and also DP-like cells (Fig. 6D, SI Fig. 6C).

In addition, some of the cluster names are not very informative, in particular "Twist1high". Without such details it is impossible to meaningfully assess the validity of the conclusions. We now use cell markers (*Cldn1*+ cells, DP-like cells) or cell conditions (SM22-Tom, SM22-SmoM2/Tom) to name the cell clusters.

3. Figure 6B: The authors state that the genes presented in this panel are expressed in the transitional cell population. In my opinion the figure does not convincingly support this statement. At least, they seem to be expressed at equal or higher levels in several other cell clusters. In addition, two of the genes (*Vcan* and *Crabp1*) are not discussed in the text and their significance is unclear. The cluster-related differences would be easier to interpret from e.g. a violin plot.

We acknowledge the weakness. Our new analysis has avoided such an ambiguous cell cluster. We also have adopted violin plots to illustrate the expression profile (Fig. 6E, SI Fig. 6C, F).

Figure 6D-J: In my opinion, the relation of these results and those of the scRNA-seq analysis is questionable, as the expression of *Twist1* and *Snai1* at RNA-level does not seem to be limited to any single cluster. Therefore, the connection of the cells studied by IHC and those analysed by scRNA-seq remains rather vague.

Twist1 is known to also function in dermal cells. Originally, we intended to focus on the epidermal cells that express *Twist1*, but it is true the background is high. Hence, we focused on another newly emerged cell cluster *Smo*+Fb from the SM22-SmoM2/Tom Fb, amongst which

the cells can be exclusively identified depending on their expression of *Cldn1*, *CD200*, and *Prlr* (Fig. 6C-E). Furthermore, IHC of WT PWD14 wounds show some of the epithelial placode cells are *SM22+*, *Cldn1+*, and *CD200+* (Fig. 6G), which suggests they could be part of the new *Smo+* populations that contribute to WIHN. As stated in response to your question 1, this is not rigorously proven by *Cre-loxP* yet. We state it is speculative in the text and are open to the possibility to remove this part.

4. Figure 6 and S. Fig 5: The lack of overlap between control and PWD14 fibroblasts is striking, and warrants more thorough analysis. The authors mention that *Grem1* and *Inhba* are differentially expressed, but the expression of these genes (or other top hits) is not shown at single-cell level. Are they ubiquitously expressed or upregulated in a specific sub-population? Furthermore, it would be important to account for the role of technical effects: were the samples processed in the same batch or at different times? Is the technical quality similar?

Our new analysis focused on DP markers (*Prlr*, *Wif1*, etc.) identified from scRNA-seq analysis of WIHN using wild type mice (Fig. 6A). We also identify the DP-like cell clusters in the newly identified *Smo+* cell cluster from *SM22-SmoM2/Tom Fb*, which is distinct from the overlapping cluster with *SM22-Tom Fb* (Fig. 6C). These are independent experiments; however, we believe the markers identified are adaptable to Lim et al.'s datasets. The technical details are explained in materials and methods.

5. Methods section:

- How many cells were sequenced from each condition?
- How deeply were the cells sequenced? How many genes were detected per cell?
 - We sequenced 2,801 cells from unwounded skin, with a median of 3,777 reads per cell; a mean of 932 genes were detected per cell.

• How was the data normalized? What units of gene expression are presented in 6B-C?

The data was normalized using the Counts per million (CPM) method. Thereafter a value of 1 was added to each data point and the values were log transformed. The gene expression unit is CPM.

- Trajectory analysis is mentioned but it is not shown in the results section
 - No trajectory analysis is used in our analysis and the main text is revised accordingly.

Reviewers' Comments:

Reviewer #1:

Remarks to the Author:

The manuscript is improved with the additional data. Because the Twist story remains, better characterization of Twist expression would help to define its role. Since Twist is absent from E14 hair placodes, its expression in WIHN placodes should be better documented. The panels in Fig 3F and 4E are not very convincing, especially when compared to the other placode marker staining. In Fig 3F it looks like Twist is expressed outside the hair placode in the inter follicular epidermis. This begs the question of when and where Twist is first expressed in the wound epithelium. Is it only expressed in the center of the wound?

The EMT conclusions are not very clear, and the reinterpretation of the Lim data does not seem to add much. What is the connection between the Lim paper and wound stiffness or Twist1? The authors claim that they found Cldn1 (an alleged epi marker) and CD200 expression in the SM22-SmoM2/Tom dermal fibroblasts. They used this as evidence of EMT. However they also claimed there were SM22 expressing epidermal cells, particularly in the placode, as they showed with IF staining in Fig 6G. How do they know that the Smo+Cldn+ cells are not epidermal cells? They also identified Prlr from unwounded skin as a dermal papilla marker. They tried to show that Prlr+ cells decreased in the K14-Cre-Twist1 mice at PWD14. Are there dermal papilla at PWD14? Is Prlr expressed in dermal condensate? What does the wide-spread expression of Prlr in the dermal fibroblasts mean as shown in Fig 6G?

Overall, the tension studies remain fascinating and the proposed mechanisms intriguing.

Reviewer #2:

Remarks to the Author:

The manuscript has been quite extensively altered in response to my comments and is now clearer and more internally consistent. I have no further comments and am happy for this to be accepted.

Reviewer #3:

Remarks to the Author:

In my opinion, this is a very interesting and innovative manuscript and it addresses an important and timely question. The authors convincingly demonstrate that hair follicle regeneration is influenced by mechanochemical signal transduction. This is shown elegantly using common lab mice and African spiny mice and further corroborated by perturbation (blebbistatin).

In the second part of the manuscript, Twist1 is identified as a potential regulator of WIHN by bulk RNA-seq. While Twist1 expression is not limited to the hair placodes and the observed fold change difference is not among the very top genes, its role of Twist1 in hair follicle regeneration is quite solidly established by inhibition and knockdown assays. However, the precise mechanistic role of Twist1 in this process cannot be concluded from the data, although different possibilities are presented in the discussion section.

Upon the initial submission, I raised some concerns regarding the scRNA-seq analyses included in the manuscript. In the revised manuscript the authors have addressed many of these and this part has been quite thoroughly re-written. Regrettably, I still think that the dataset has not been analyzed to full potential. The analyses focus on selected rather complex comparisons, and in my opinion the presented results do not offer significant new insights into the role of Twist1. I agree with the authors that this part is not essential to the manuscript and the story. Unless the authors decide to omit this part and save it for their next paper, I would recommend addressing the following issues:

-The expression of Twist1 is not shown at single-cell level. Neither are the genes that it is proposed to regulate (in Fig 3E). In my opinion these analyses would be logical given the centrality of Twist1 in this manuscript. While it may be that Twist1 expression does not show significant differences between the cell clusters, in my opinion this would still be important to discuss.

-Figure 6C-E: It is difficult to understand how exactly these comparisons have been performed. What is the relationship of the clusters in E and the tSNE plot in C? To which cells exactly were the DP-like cells compared? The populations presented in E should be shown in tSNE space either in Fig. 6 or in supplements.

-Besides the identification of DP-like cells, the heterogeneity of fibroblasts is not discussed. However, the tSNE plots (6A and C) suggest that several subpopulations or states might exist?
- While the method section is now much improved, it is still not clear what clustering method was used (for example K-means clustering or graph-based clustering?). t-SNE is a dimensionality reduction technique, not a clustering method. This is a relatively minor detail, but I would still recommend correcting this.

-Figure 4B-D: How can CPM values be negative? Have the values been transformed and/or scaled?

-Figure 5 legend, line 1023: Should be E-G, not D-F?

REVIEWER COMMENTS

Reviewer #1 (Remarks to the Author):

1. The manuscript is improved with the additional data. Because the Twist story remains, better characterization of Twist expression would help to define its role. Since Twist is absent from E14 hair placodes, its expression in WIHN placodes should be better documented. The panels in Fig 3F and 4E are not very convincing, especially when compared to the other placode marker staining. In Fig 3F it looks like Twist is expressed outside the hair placode in the inter follicular epidermis. This begs the question of when and where Twist is first expressed in the wound epithelium. Is it only expressed in the center of the wound?

Thank you for the constructive comment and opportunity to clarify. Please see below for our explanations. The “quoted text” are included in the revised manuscript.

After wounding, there is a wound bed (red) whose fate have not been specified. “We propose that there are two levels of symmetry breaking during successful wound induced hair neogenesis (WIHN). The first level is the generation of morphogenesis competent field (green) from the center. The second level is the generation of periodically arranged hair germs forming (yellow dots) from the morphogenesis competent field (Left column of the following figure. Panels are from Fig. 6). In the spiny mice, the topology is reversed with the competence zone (green) on the periphery, while the central field (blue) cannot form hairs (Right column). The key question is what makes a region competent to undergo further periodic Turing patterning to generate hair placodes” (Line 384-395). Certainly, many morphogens (Wnt, Shh, FGF9, etc. cited in the manuscript) are involved. In our study, we focus on the role of tissue mechanics, and found Twist1 to play a pivotal role in this competence. However, Twist1 is enriched in the morphogenetic zone (green zone), but not limited to the hair placode (yellow dots). How do we explain this?

“Based on our earlier developmental studies of periodic formation of the feather and hair germs, we have noticed there are two waves of molecular expression, which we name them restrictive and *de novo* mode, respectively (Chuong et al., 2013, and part of its Fig. 3 is shown below)” (Line 493-501).

In parallel to the description for WIHN above, the skin field (white, middle column in figure below) originally are not capable of undergoing periodic patterning. First, there is the formation of the morphogenesis competent field (light blue). Second, Turing patterning takes place within the morphogenetic competent field to generate periodically arranged buds (dark blue), which creates a lateral inhibitory zone (white) surrounding each bud.

Here we use β -Catenin, Shh and phosphorylated ERK as examples. “Restrictive mode” expression is represented by β -Catenin (left column) with a more or less homogeneous expression in the whole field. When new buds form (represented by Shh expression), which is in *de novo* mode, Shh is expressed at the center of the placode. Therefore, restrictive mode molecules (e.g., β -Catenin) are more important as their presence allow Turing patterning to occur, while *de novo* mode molecules (e.g., Shh) are the consequence of the Turing patterning process. Phosphorylated ERK is another restrictive mode molecular event, and its expression during bud formation is converted from a general expression pattern (right column, upper panel, above the midline) to distinct bud pattern (right column, lower panel). Cellular events of the formation of the competent field and the periodic patterning process are shown schematically in the bottom panel.

To come back to this study, it has been known that hair germ formation in WIHN may not use the exact same molecular pathway as in embryonic development, albeit they both achieve final formation of hair placodes. We are aware that Twist1 is not expressed in the epidermis of developing skin. We are surprised to see epithelial Twist1 expression associated with the morphogenetic field in both laboratory and spiny mice. Expression data and our experiments in which softening the wound bed or overexpressing Twist1 in the wound enhanced WIHN (Fig. 5) demonstrate that it belongs to the “restrictive mode” molecules, molecules that are present before periodic patterning occurs, and therefore they are present in the epidermis in both putative placode and inter-placode regions. We now present a new figure panel to show the confocal imaging of Twist1 in three different regions of the wound bud (Fig. 3F and below). The staining is reduced to near absence towards the periphery. Indeed, we can see Twist1 is present in both placode and interplacode region in both laboratory mice (Fig. 3F, G) and spiny mice (Fig. 4E). The spiny mice skin has more Twist1 expression in the dermis than epidermis. As for Twist1 expression in the dermis, since they are highly expressed without significant differences in central or peripheral wound fields (Supplement Fig. 3C), we do not consider they play a critical role in determining the outcome of WIHN.

New Fig. 3F

Restrictive mode molecules are required for periodic patterning process, while *de novo* mode is the readout of the periodic patterning process. Based on this study, “our view is that **Twist1 is initially broadly expressed in the morphogenetic zone of the wound bed, and become accentuated in the placode region and enter the nucleus when the sum of all the upregulating factors for placode formation reaches a threshold**”. In the inter-placode region, Twist1 will be turned off. Although we show that epidermal Twist1 is essential for wound-induced hair placode formation, other mechanical and chemical factors are also required for the cells to enter a competent stage. Lastly, while the focus of this study is about tissue mechanics, WIHN process is a combination of local periodic patterning events and a global influence that constitute the morphogenetic field. The related discussion is included in Line 499-505.

Module-based complexity formation: periodic patterning in feathers and hairs.

Chuong CM, Yeh CY, Jiang TX, Widelitz R.

Wiley Interdiscip Rev Dev Biol. 2013 Jan-Feb;2(1):97-112.

2. The EMT conclusions are not very clear, and the reinterpretation of the Lim data does not seem to add much. What is the connection between the Lim paper and wound stiffness or Twist1? The authors claim that they found Cldn1 (an alleged epi marker) and CD200 expression in the SM22-SmoM2/Tom dermal fibroblasts. They used this as evidence of EMT. However they also claimed there were SM22 expressing epidermal cells, particularly in the placode, as they showed with IF staining in Fig 6G. How do they know that the Smo+Cldn+ cells are not epidermal cells? They also identified Prlr from unwounded skin as a dermal papilla marker. They tried to show that Prlr+ cells decreased in the K14-Cre-Twist1 mice at PWD14. Are there dermal papilla at PWD14? Is Prlr expressed in dermal condensate? What does the wide-spread expression of Prlr in the dermal fibroblasts mean as shown in Fig 6G?

In the response to Question 1, we explained more clearly that Twist1 is one of the restrictive mode molecules that make morphogenetic field competent to undergo further periodic patterning process. Yet, how does Twist1 function in WIHN? It is our aspiration to find out the mechanism of this epidermal Twist 1 expression that led us to pursue scRNA analyses and study the fate of Twist1 positive cell cluster. However, we must acknowledge that, without a comprehensive scRNA analyses and verifications such as cell lineage study (e.g., the use of Twist1 promoter-driven Cre to cross with a reporter mouse), the proposed downstream EMT events cannot be concluded. We do appreciate Reviewer 1 for summarizing his/her attitude towards this work as “Overall, the tension studies remain fascinating and the proposed mechanisms intriguing.”

Reviewer 3 has similar comments. He/she raises several issues about scRNA analyses and states “the precise mechanistic role of Twist1 in this process cannot be concluded from the data”. Yet, he/she also states that “The authors convincingly demonstrate that hair follicle regeneration is influenced by mechanochemical signal transduction” and agrees that “this (scRNA) part is not essential to the manuscript and the story.”

We appreciate all reviewers' very helpful scientific questions and the constructive attitude to let the tissue mechanics/WIHN part of the paper go forward and put unsolved issues in discussion. Hence, we have decided to remove the scRNA data from the current manuscript. We have clarified our thoughts on how Twist1 may work in discussion, but have left the mechanism as unknown, which warrants further studies. We will investigate deeper into the mechanistic regulation of wound stiffness and Twist1, and to refine these data in conjunction with additional genetic and single cell studies for inclusion in a future manuscript. The related discussion is addressed in Line:506-510.

Reviewer #2 (Remarks to the Author):

1. The manuscript has been quite extensively altered in response to my comments and is now

clearer and more internally consistent. I have no further comments and am happy for this to be accepted.

We appreciate the constructive comments that have improved this manuscript.

Reviewer #3 (Remarks to the Author):

1. In my opinion, this is a very interesting and innovative manuscript and it addresses an important and timely question. The authors convincingly demonstrate that hair follicle regeneration is influenced by mechanochemical signal transduction. This is shown elegantly using common lab mice and African spiny mice and further corroborated by perturbation (blebbistatin).

In the second part of the manuscript, Twist1 is identified as a potential regulator of WIHN by bulk RNA-seq. While Twist1 expression is not limited to the hair placodes and the observed fold change difference is not among the very top genes, its role of Twist1 in hair follicle regeneration is quite solidly established by inhibition and knockdown assays. However, although different possibilities are presented in the discussion section.

Thank you. Please kindly see our response to point 1, reviewer 1 in which we explained our thoughts on why the expression of Twist 1 is not limited to hair placodes.

2. Upon the initial submission, I raised some concerns regarding the scRNA-seq analyses included in the manuscript. In the revised manuscript the authors have addressed many of these and this part has been quite thoroughly re-written. Regrettably, I still think that the dataset has not been analyzed to full potential. The analyses focus on selected rather complex comparisons, and in my opinion the presented results do not offer significant new insights into the role of Twist1. I agree with the authors that this part is not essential to the manuscript and the story. Unless the authors decide to omit this part and save it for their next paper, I would recommend addressing the following issues:

-The expression of Twist1 is not shown at single-cell level. Neither are the genes that it is proposed to regulate (in Fig 3E). In my opinion these analyses would be logical given the centrality of Twist1 in this manuscript. While it may be that Twist1 expression does not show significant differences between the cell clusters, in my opinion this would still be important to discuss.

-Figure 6C-E: It is difficult to understand how exactly these comparisons have been performed. What is the relationship of the clusters in E and the tSNE plot in C? To which cells exactly were the DP-like cells compared? The populations presented in E should be shown in tSNE space either in Fig. 6 or in supplements.

-Besides the identification of DP-like cells, the heterogeneity of fibroblasts is not discussed. However, the tSNE plots (6A and C) suggest that several subpopulations or states might exist?
- While the method section is now much improved, it is still not clear what clustering method was used (for example K-means clustering or graph-based clustering?). t-SNE is a

dimensionality reduction technique, not a clustering method. This is a relatively minor detail, but I would still recommend correcting this.

Thanks. We truly appreciate your candid comments and excellent advices.

Please kindly see our response to point 2, reviewer 1, about our agreement to delete scRNA data and to pursue a more comprehensive study to delineate the mechanistic role of Twist1 in WIHN in the future.

3. Figure 4B-D: How can CPM values be negative? Have the values been transformed and/or scaled?

Thank you. The previous unit was calculated based on the original variable, minus its mean, divided by its standard deviation. We have revised the label to z-score.

4. Figure 5 legend, line 1023: Should be E-G, not D-F?

Sorry for the typo. They are corrected.

Reviewers' Comments:

Reviewer #1:

Remarks to the Author:

The new figure, showing that Twist 1 was enhanced in the wound center epidermis, and further concentrated in the placodes, but diminished towards wound edge is an important piece of data, supporting that epidermal Twist 1 could play an important role in neogenic hair formation. Though the twist story is not the strongest part of the manuscript, the overall impact of the findings are high.